# Bridging the Gap between ANNs and SNNs by Calibrating Offset Spikes

**Zecheng Hao**[1]**, Jianhao Ding**[1]**, Tong Bu**[1,2]**, Tiejun Huang**[1,2] **& Zhaofei Yu**[1,2] [*]
[1] School of Computer Science, Peking University
[2] Institute for Artificial Intelligence, Peking University

## Abstract

Spiking Neural Networks (SNNs) have attracted great attention due to their distinctive characteristics of low power consumption and temporal information processing. ANN-SNN conversion, as the most commonly used training method for applying SNNs, can ensure that converted SNNs achieve comparable performance to ANNs on large-scale datasets. However, the performance degrades severely under low quantities of time-steps, which hampers the practical applications of SNNs to neuromorphic chips. In this paper, instead of evaluating different conversion errors and then eliminating these errors, we define an offset spike to measure the degree of deviation between actual and desired SNN firing rates. We perform a detailed analysis of offset spike and note that the firing of one additional (or one less) spike is the main cause of conversion errors. Based on this, we propose an optimization strategy based on shifting the initial membrane potential and we theoretically prove the corresponding optimal shifting distance for calibrating the spike. In addition, we also note that our method has a unique iterative property that enables further reduction of conversion errors. The experimental results show that our proposed method achieves state-of-the-art performance on CIFAR-10, CIFAR-100, and ImageNet datasets. For example, we reach a top-1 accuracy of 67.12% on ImageNet when using 6 time-steps. To the best of our knowledge, this is the first time an ANN-SNN conversion has been shown to simultaneously achieve high accuracy and ultralow latency on complex datasets. Code is available at https://github.com/hzc1208/ANN2SNN_COS.

## 1 Introduction

Acclaimed as the third generation of Artificial Neural Networks (Maass, 1997), Spiking Neural Networks (SNNs) have brought brand-new inspiration to computational neuroscience. As the corresponding neuron fires spikes only when the current membrane potential exceeds the firing threshold, SNNs have the distinctive characteristics of binary output, high sparsity, and biological plausibility. Therefore, compared with traditional ANN models, SNNs can further improve computational efficiency and reduce power consumption, which facilitates their remarkable superiority in the application of neuromorphic chips (Merolla et al., 2014; Davies et al., 2018; DeBole et al., 2019). Considering that an effective learning algorithm has not yet been found for SNNs, ANN-SNN conversion and backpropagation through time (BPTT) are still the two most commonly applied training methods. Compared with BPTT, ANN-SNN conversion provides a way around the nondifferentiable problem in the direct training procedure for SNNs and thus reduces the overall training complexity.

The aim in ANN-SNN conversion is to establish the mapping relationship between the activation output and the average firing rate. Traditional conversion methods exploit larger time-steps to overcome conversion errors and thus achieve high performance (Diehl et al., 2015). Many of the following works have attempted to optimize the performance from multiple perspectives, including using the soft-reset mechanism (Han et al., 2020), proposing more adaptive activation functions (Ho & Chang, 2021; Bu et al., 2022b), adopting a trainable threshold (Sengupta et al., 2019; Ding et al., 2021; Bu et al., 2022a), etc. However, these strategies cannot effectively eliminate the errors caused

---

[*]Corresponding author: yuzf12@pku.edu.cn

by the deviation between the actual and desired firing rates, especially when the number of time-steps is small. Some recent works explore compensating for the errors by introducing burst spikes (Li & Zeng, 2022) and signed spiking neurons (Li et al., 2022). Unlike these works, our paper attempts to eliminate the errors with vanilla spiking neurons and answer the question of how to improve the performance of a converted SNN and possibly approach the upper bound performance.

In this paper, we observe and identify the source of conversion errors and propose an iterative optimization method based on shifting the initial membrane potential, which can fulfil accurate mapping between ANNs and SNNs under ideal situations. Our main contributions are summarized as follows:

1 We introduce the concept of offset spike to infer the deviation between the actual SNN firing rate and the desired SNN firing rate. We note that cases of firing one additional (or one less) spike are the main reason cause of conversion errors.

2 We propose a method to judge offset spike based on the residual membrane potential and an optimization method to eliminate conversion errors by shifting the initial membrane potential up or down. We derive the optimal shifting distance and prove that one spike can be increased or decreased under this condition.

3 We evaluate our methods on CIFAR-10/100 and ImageNet datasets. The proposed method outperforms the existing state-of-the-art ANN-SNN conversion methods using fewer time-steps. For example, we achieve 67.12% top-1 accuracy on ImageNet with only 6 time-steps (4 time-steps for calibration and 2 time-steps for inference). Moreover, it is worth noting that we have reached the same level of performance as BPTT under the condition of significantly reduced memory and computing resources requirements.

4 We discover that our proposed method has an iterative property. Under ideal circumstances, the deviation within the range of $k$ spikes will be eliminated entirely after adopting our approach $k$ times. After 4 iterations, the mean-square error between the actual and desired firing rates of the output layer can reach 0.001 for the VGG-16 model on CIFAR-100.

## 2 RELATED WORKS

The principle of ANN-SNN conversion is to map the parameters from pretrained ANN models to SNNs, which avoids training SNNs directly and reduces energy consumption significantly. The primary goal is to match the ANN activation value and the average SNN firing rate. Cao et al. (2015) were the research pioneers in this field, and they replaced ReLU activation layers in ANNs with spiking neurons to fulfil the conversion procedure. Ho & Chang (2021); Bu et al. (2022b) proposed new activation functions, which better fit the finiteness and discreteness of the spike firing rate. Rueckauer et al. (2017); Han et al. (2020) adopted "reset-by-subtraction" mechanism, which alleviated the problem of information loss and effectively improved the precision of the conversion process. For the setting of firing threshold, various strategies have been proposed, including RobustNorm (Rueckauer et al., 2017), SpikeNorm (Sengupta et al., 2019), and adjustable threshold (Han et al., 2020; Ding et al., 2021; Ho & Chang, 2021; Bu et al., 2022a;b), etc. Recently, spiking neural networks with high accuracy and low latency have become the focus and target of academic research. To reduce the time latency of the network, one must carefully address the exact spiking time of neurons. Deng & Gu (2021); Li et al. (2021) fine-tuned the bias in each layer under the uniform current assumption. Nevertheless, the actual current would never be distributed uniformly. In terms of expectation, Bu et al. (2022b) proved that one-half of the threshold is the optimal value for the initial membrane potential, and that charging at this value can prompt neurons to spike more uniformly. However, as the authors pointed out: there is still a mismatch between ANN and SNN due to the so called "unevenness error". In addition, other methods like burst spikes (Li & Zeng, 2022) and signed spiking neurons (Wang et al., 2022a; Li et al., 2022), have also been introduced to further improve performance. These efforts have aimed to alleviate the conversion loss. However, they undermined the biological plausibility and binary property of spiking neurons.

In addition to ANN-SNN conversion, backpropagation with exact spike time is another common way to train SNNs. The surrogate gradient (O'Connor et al., 2018; Zenke & Ganguli, 2018; Bellec et al., 2018; Wu et al., 2018; 2019; Kim & Panda, 2020; Zenke & Vogels, 2021) has been widely used to tackle the nondifferentiable problem in the training process, which substitutes the Heaviside function with a derivable function. With the help of the surrogate gradient, backpropagation

through time enables the network to adjust weights by focusing every exact time-step. On this basis, Rathi & Roy (2021); Guo et al. (2022) further attempted the optimization of hyper-parameters and gradients. Bohte et al. (2002); Kheradpisheh & Masquelier (2020); Zhang & Li (2020) proposed a timing-based learning method, which viewed the specific spike firing time as significant temporal information to transmit between layers. Nevertheless, this type of method only applies to shallow networks at present. In addition, hybrid training methods have recently attracted extensive attention. Wang et al. (2022b); Rathi & Roy (2021) combined ANN-SNN conversion with BPTT to obtain higher performance under low latency. Kim et al. (2020) adopted rate-coding and time-coding simultaneously to train SNNs with fewer spikes. Mostafa (2017); Zhou et al. (2021); Zhang & Li (2020) established a linear transformation about the spike firing time from adjacent layers, which enabled the use of SNNs under the training mode of ANNs. In addition, BPTT can enable calibration of the spike time in the training phase (Rathi et al., 2020). These works alter weights when training and stress the importance of spike timing, which is usually ignored in conversion methods. Inspired by these approaches, we incorporate the concept of calibration spike timing by manipulating membrane potentials into the conversion pipeline to bridge the gap between ANNs and SNNs.

## 3 PRELIMINARIES

### 3.1 NEURON MODELS

For ANNs, the input $\boldsymbol{a}^{l-1}$ to layer $l$ is mapped to the output $\boldsymbol{a}^l$ by a linear transformation matrix $\boldsymbol{W}^l$ and a nonlinear activation function $f(\cdot)$, that is ($l = 1, 2, 3, \cdots, L$):

$$\boldsymbol{a}^l = f(\boldsymbol{W}^l \boldsymbol{a}^{l-1}). \tag{1}$$

where $f(\cdot)$ is often set as the ReLU activation function.

For SNNs, we adopt Integrate-and-Fire (IF) Neuron model (Gerstner & Kistler, 2002), which is similar to the approach reported in previous works (Cao et al., 2015; Diehl et al., 2015). To minimize information loss during inference, our neurons perform "reset-by-subtraction" mechanism (Han et al., 2020), which means that the firing threshold $\theta^l$ is subtracted from the membrane potential after firing. The overall kinetic equations of IF Neuron can be expressed as follows:

$$\boldsymbol{v}^l(t) = \boldsymbol{v}^l(t-1) + \boldsymbol{I}^l(t) - \boldsymbol{s}^l(t)\theta^l, \tag{2}$$

$$\boldsymbol{I}^l(t) = \boldsymbol{W}^l \boldsymbol{s}^{l-1}(t)\theta^{l-1}. \tag{3}$$

Here $\boldsymbol{v}^l(t)$ and $\boldsymbol{I}^l(t)$ denote the membrane potential and input current of layer $l$ at the $t$-th time-step, respectively. $\boldsymbol{W}^l$ is the synaptic weight between layer $l - 1$ and layer $l$, and $\theta^l$ is the spike firing threshold in the $l$-th layer. $\boldsymbol{s}^l(t)$ represents whether the spike fires at time-step $t$. For the $i$-th neuron, if the current potential exceeds the firing threshold $\theta^l$, the neuron will emit a spike. This firing rule can be described by the equation below.

$$s_i^l(t) = \begin{cases} 1, & v_i^l(t-1) + I_i^l(t) \geqslant \theta^l \\ 0, & v_i^l(t-1) + I_i^l(t) < \theta^l \end{cases}. \tag{4}$$

If not otherwise specified, the subscript $x_i$ denotes the $i$-th element of $\boldsymbol{x}$.

### 3.2 ANN-SNN CONVERSION

The main principle of ANN-SNN conversion is to map the firing rates (or postsynaptic potential) of spiking neurons to the ReLU activation output of artificial neurons. Specifically, by summing equation 2 from $t = 1$ to $t = T$, and then substituting variable $\boldsymbol{I}^l(t)$ with $\boldsymbol{W}^l \boldsymbol{s}^{l-1}(t)\theta^{l-1}$ using equation 3, and finally dividing $T$ on both sides, we obtain the following equation:

$$\frac{\sum_{t=1}^{T} \boldsymbol{s}^l(t)\theta^l}{T} = \boldsymbol{W}^l \frac{\sum_{t=1}^{T} \boldsymbol{s}^{l-1}(t)\theta^{l-1}}{T} + \left( -\frac{\boldsymbol{v}^l(T) - \boldsymbol{v}^l(0)}{T} \right). \tag{5}$$

where $T$ denotes the total simulation cycle. For simplicity, we use the average postsynaptic potential $\boldsymbol{\phi}^l(T)$ as a substitute for the term $\sum_{t=1}^{T} \boldsymbol{s}^l(t)\theta^l / T$ in equation 5, then we obtain

$$\boldsymbol{\phi}^l(T) = \boldsymbol{W}^l \boldsymbol{\phi}^{l-1}(T) + \left( -\frac{\boldsymbol{v}^l(T) - \boldsymbol{v}^l(0)}{T} \right). \tag{6}$$

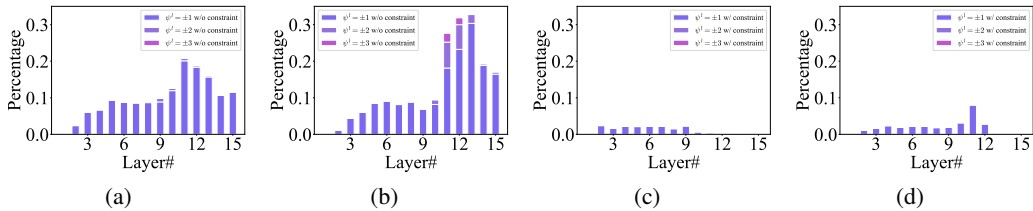

Figure 1: The distribution of offset spike in each layer. (a) and (c): VGG-16 on CIFAR-10, (b) and (d): VGG-16 on CIFAR-100. w/ constraint denotes the constraint of $\boldsymbol{\psi}^{l-1} = 0$ through the rectification of the spikes.

Equation 6 can be approximated by a linear transformation between $\phi^l(T)$ and $\phi^{l-1}(T)$ as $T$ tends to infinity, which is exactly the same as the forward propagation (equation 1) in ANNs due to $\phi^l(T) \geqslant 0$. This result implies that we can achieve lossless ANN-SNN conversion when $T$ tends to infinity. However, the performance of converted SNNs degrades seriously under the condition of short time-steps $T$ (Rueckauer et al., 2017; Han et al., 2020). To achieve high-performance SNNs under low latency, Bu et al. (2022b) proposed replacing the commonly used ReLU activation function of source ANNs with the quantization clip-floor-shift (QCFS) function:

$$\boldsymbol{a}^l = f(\boldsymbol{a}^{l-1}) = \frac{\lambda^l}{L}\text{clip}\left(\left\lfloor \frac{\boldsymbol{W}^l \boldsymbol{a}^{l-1} L}{\lambda^l} + \frac{1}{2}\right\rfloor, 0, L\right). \tag{7}$$

where $L$ denotes the ANN quantization step and $\lambda^l$ is the trainable threshold of the outputs in ANN layer $l$, which is mapped to the threshold $\theta^l$ in SNN layer $l$. This paper follows the conversion framework of Bu et al. (2022b) with QCFS function.

## 4 METHODS

In this section, we first compare the outputs of ANNs and converted SNNs in each layer. We introduce offset spike to measure the degree of deviation between the actual firing rate and the desired firing rate in SNNs. Then, we demonstrate that the offset spike of being one accounts for the main part in each layer and is the main reason of conversion error. Based on this, we propose sufficient conditions to determine if offset spike exists and what the sign of offset spike value is, and we present a spike calibration strategy to eliminate conversion errors through shifting the initial membrane potential.

### 4.1 OFFSET SPIKE AND ITS DISTRIBUTION

ANN-SNN conversion errors can be divided into clipping error, quantization error (flooring error), and unevenness error (deviation error) (Bu et al., 2022b). In previous works (Han et al., 2020; Li et al., 2021; Meng et al., 2022b), those errors are eliminated (or reduced) separately, and thus far no method to eliminate the unevenness error (deviation error) has been identified. Since we find that the essential cause of most conversion errors comes from the remaining term $-\frac{\boldsymbol{v}^l(T) - \boldsymbol{v}^l(0)}{T}$ in equation 5, we consider reducing conversion errors directly based on the prior knowledge of the remaining term. To measure the degree of deviation between the actual firing rate and the desired firing rate, we first introduce the definition of offset spike.

**Definition 1.** We define **OFFSET SPIKE** $\boldsymbol{\psi}^l$ of layer $l$ as the difference between the desired total spike count $C^l_{\text{designed}}$ and the actual spike count $C^l_{\text{actual}}$ during the interval $[0, T]$, that is

$$\boldsymbol{\psi}^l = C^l_{\text{designed}} - C^l_{\text{actual}} = \frac{\boldsymbol{a}^l T}{\theta^l} - \sum_{t=1}^{T} \boldsymbol{s}^l(t). \tag{8}$$

where we set the maximum value $\lambda^l$ of output $\boldsymbol{a}^l$ in ANNs equal to the threshold $\theta^l$ in SNNs, that is, $\lambda^l = \theta^l$. Thus, $\frac{\boldsymbol{a}^l}{\theta^l}$ denotes the normalized output in ANNs, which is mapped to the firing rates of

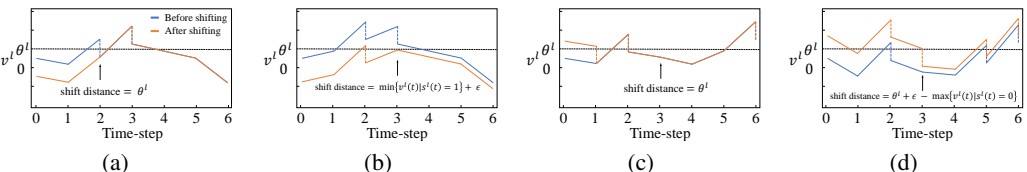

Figure 2: Shifting up (down) the initial membrane potential can increase (decrease) one output spike.

SNNs, and $C_{\text{designed}}^l = \frac{a^l T}{\theta^l}$ denotes the desired total spike count. Note that $\psi_i^l = \pm k$ indicates that the gap between the actual and desired firing rate of the $i$-th neuron in layer $l$ of the SNN is $k$ spikes. We further investigate the detailed ANN and SNN outputs in each layer. We train the source ANN with the QCFS activation function (equation 7) and then convert it to an SNN (more details are in the Appendix). Fig. 1(a)-1(b) illustrates the distribution of offset spike for the converted SNNs with VGG-16 structure on CIFAR-10 and CIFAR-100, respectively. We have the following observation.

**Observation 1.** $\psi^l = \pm 1$ accounts for the main part in each layer and $\psi^l = \pm 3$ rarely occurs.

Considering the cumulative effect of conversion errors in the deep layer, the offset spike $\psi^l$ in layer $l$ can be considered as the joint effects of the offset spike $\psi^{l-1}$ in layer $l-1$ and conversion errors in layer $l$, and tends to increase with the increase in the number of layers.

For a deeper analysis of the offset spike in each layer, we rectify ANN output in layer $l-1$ to make $a^{l-1} = \phi^{l-1}(T)$ and $\psi^{l-1} = 0$ (Sec. A.1 for more details of the constraint), and then compute the offset spike $\psi^l$ in layer $l$. After the rectification for each layer, the distribution of offset spike for the converted SNNs with VGG-16 structure on CIFAR-10 and CIFAR-100 are shown in Fig. 1(c)-1(d), respectively. We have the following observation.

**Observation 2.** With constraint, $\psi^l = \pm 1$ accounts for the main part in each layer and $|\psi^l| > 1$ rarely occurs.

Observations 1-2 show that the firing of one additional (or one less) spike is the main cause of conversion errors, which implies that we can eliminate errors after adjusting $\sum_{t=1}^{T} s^l(t)$ with $\pm 1$.

## 4.2 JUDGE CONVERSION ERRORS THROUGH RESIDUAL MEMBRANE POTENTIAL

Before we propose the optimal strategy for adjusting the output spikes to eliminate the offset spike $\psi^l$, we need to determine if the offset spike exists and what the sign of the offset spike value is. If the sign of the offset spike is positive, which corresponds to the situation in which the ANN output is larger than the SNN output, the spiking neurons should fire more spikes to eliminate the offset spike, otherwise, they should fire fewer spikes.

In the practical application of SNNs, we cannot directly obtain the specific value of the offset spike $\psi^l$. Fortunately, we find that we can determine the sign of $\psi^l$ according to the value of the residual membrane potential $v^l(T)$. We have the following theorem:

**Theorem 1.** *Suppose that an ANN with QCFS activation function (equation 7) is converted to an SNN with $L = T, \lambda^l = \theta^l, v^l(0) = \theta^l/2$, and the inputs to the $l$-th layer of the ANN and the SNN are the same, that is, $a^{l-1} = \phi^{l-1}(T)$. Then for any $i$-th element of the $l$-th layer, we can draw the following conclusions:*
*(i) If $\phi_i^l(T) > 0$ and $v_i^l(T) < 0$, we will have $\phi_i^l(T) > a_i^l$ and $\psi_i^l < 0$.*
*(ii) If $\phi_i^l(T) < \theta^l$ and $v_i^l(T) \geqslant \theta^l$, we will have $\phi_i^l(T) < a_i^l$ and $\psi_i^l > 0$.*

The proof is provided in Appendix. (i) implies that if the postsynaptic potential is larger than 0 and the residual membrane potential is smaller than 0, we can conclude that the neuron fires more spikes than expected and the sign of the offset spike value is negative. (ii) implies that if the postsynaptic potential is smaller than 0 and the residual potential is larger than $\theta$, we can conclude that the spiking neuron fires fewer spikes than expected and the sign of the offset spike value is positive.

### 4.3 ELIMINATE CONVERSION ERROR THROUGH SHIFTING INITIAL MEMBRANE POTENTIAL

Since the firing of one additional (or one less) spike ($\psi^l = \pm 1$) is the main cause of conversion errors, we propose an optimization strategy to rectify the output spike $\sum_{t=1}^{T} s^l(t)$ by adding or subtracting one spike, thereby eliminating errors. Specifically, we consider adjusting the value of $\sum_{t=1}^{T} s^l(t)$ by shifting the corresponding initial membrane potential $v^l(0)$ up or down. One intuitive explanation is that a higher initial membrane potential will make the spiking neurons fire earlier and will increase the firing rates during the period $[0, T]$, while a lower initial membrane potential will make the spiking neurons fire later and will decrease the firing rates. The following theorem gives the optimal shifting distance when we attempt to move $\sum_{t=1}^{T} s^l(t)$ (and $\psi^l$) by a distance of $\pm 1$.

**Theorem 2.** *If we use $s_i^l(t)$ and $\widetilde{s}_i^l(t)$ to denote the binary spike of the $i$-th neuron in layer $l$ at time-step $t$ before and after optimization, $v_i^l(0)$ and $\widetilde{v}_i^l(0)$ to represent the initial membrane potential before and after optimization, then $\forall \epsilon \in (0, \theta^l)$, we will have the following conclusions:*

*(i) If we set $\widetilde{v}_i^l(0) = v_i^l(0) - \max(\theta^l, \min\{v_i^l(t)|s_i^l(t)=1\} + \epsilon)$, then $\sum_{t=1}^{T} \widetilde{s}_i^l(t) = \sum_{t=1}^{T} s_i^l(t) - 1$.*

*(ii) If we set $\widetilde{v}_i^l(0) = v_i^l(0) + \max(\theta^l, \theta^l + \epsilon - \max\{v_i^l(t)|s_i^l(t)=0\})$, then $\sum_{t=1}^{T} \widetilde{s}_i^l(t) = \sum_{t=1}^{T} s_i^l(t) + 1$.*

The proof is provided in Appendix. Note that the variable $\epsilon$ illustrates that as long as the initial membrane potential is within a certain range, the number of output spikes can be guaranteed to increase or decrease by 1.

**Example 1.** Fig. 2 shows four different scenarios before and after shifting $v^l(0)$ that verify the effectiveness of our theorem. Specifically, Fig. 2(a)-2(b)/Fig. 2(c)-2(d) indicates two cases of shifting down/up, which correspondes to (i)/(ii) in Theorem 2.

By combining Theorems 1 and 2, we propose the complete spike calibration algorithm. Our method can be divided into two stages. First, for $l$-th layer, we spend $\rho$ time-steps to determine the specific spike firing situation. According to Theorem 1, if $v_i^l(\rho) < 0$ (or $v_i^l(\rho) \geqslant \theta^l$), by combining $\phi_i^l(\rho)$, we can infer that $\phi_i^l(\rho)$ is actually larger (or smaller) than the expected average postsynaptic potential $a_i^l$, and $\psi_i^l < 0$ (or $\psi_i^l > 0$). In addition, we will preserve the membrane potential after each time-step, which will be used to calculate the subsequent optimal shifting distance.

In the second stage, based on Theorem 2, we will calculate the optimal shifting distance of the initial membrane potential for specific neurons with conversion errors. Generally speaking, if $\psi_i^l < 0$, we will shift its initial membrane potential down by calculating (i) from Theorem 2, if $\psi_i^l > 0$, we will shift its initial membrane potential up by adopting (ii) from Theorem 2. After optimizing the initial membrane potential, we will spend $T$ time-steps implementing the test on corresponding datasets and deliver the output to the $l + 1$-th layer.

### 4.4 ITERATIVE PROPERTY OF OUR OPTIMIZATION METHOD

In the previous section, we show that shifting the initial membrane potential up (or down) can change a case of $\psi^l = \pm 1$ to $\psi^l = 0$. In fact, our method also converts the case of $\psi^l = \pm k$ to $\psi^l = \pm(k-1)$. As long as the offset spike $\psi^l$ is not zero, the performance of converted SNNs will degrade. One important problem to address is whether we can further eliminate the offset spike in situations where $\psi^l \geqslant 2$.

Fortunately, we find that our optimization method has an iterative property. One can reuse Theorem 2 to increase (or decrease) one output spike each time. Of course, it comes at a significant computational cost. In the Experiments Section, we will show that the performance of the converted SNN increases with the iteration. Typically, we can achieve high-performance and low-latency SNNs with only one iteration.

## 5 EXPERIMENTS

In this section, we choose image classification datasets to validate the effectiveness and performance of our proposed methods, including CIFAR-10 (LeCun et al., 1998), CIFAR-100 (Krizhevsky et al.,

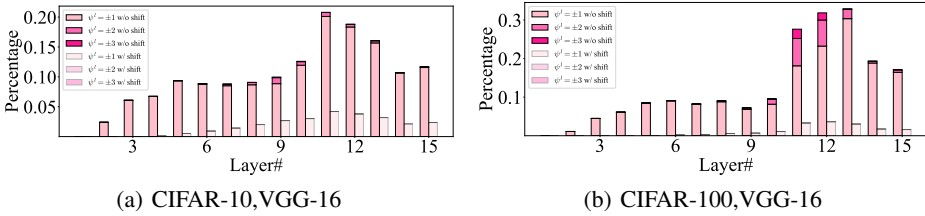

(a) CIFAR-10,VGG-16          (b) CIFAR-100,VGG-16

Figure 3: The distribution of offset spike before and after optimization

Table 1: Comparison with existing state-of-the-art ANN-SNN conversion methods

| Method | ANN | Architecture | T=1 | T=2 | T=4 | T=8 | T=16 | T=32 |
|--------|-----|--------------|-----|-----|-----|-----|------|------|
| **CIFAR-100 Dataset** | | | | | | | | |
| SNM | 74.13% | | - | - | - | - | - | 71.80% |
| SNNC-AP | 77.89% | | - | - | - | - | - | 73.55% |
| OPI | 76.31% | VGG-16 | - | - | - | 60.49% | 70.72% | 74.82% |
| QCFS | 76.28% | | - | 63.79% | 69.62% | 73.96% | 76.24% | 77.01% |
| **Ours** | 76.28% | | 74.24% | 76.03% | 76.26% | 76.52% | 76.77% | 76.96% |
| RMP | 68.72% | | - | - | - | - | - | 27.64% |
| OPI | 70.43% | ResNet-20 | - | - | - | 23.09% | 52.34% | 67.18% |
| QCFS | 69.94% | | - | 19.96% | 34.14% | 55.37% | 67.33% | 69.82% |
| **Ours** | 69.97% | | 59.22% | 64.21% | 65.18% | 67.17% | 69.44% | 70.29% |
| **ImageNet Dataset** | | | | | | | | |
| SNNC-AP | 75.36% | | - | - | - | - | - | 63.64% |
| SNM | 73.18% | | - | - | - | - | - | 64.78% |
| OPI | 74.85% | VGG-16 | - | - | - | 6.25% | 36.02% | 64.70% |
| QCFS | 74.29% | | - | - | - | 19.12% | 50.97% | 68.47% |
| **Ours** | 74.19% | | 63.84% | 70.59% | 72.94% | 73.82% | 74.09% | 74.33% |
| SNNC-AP | 75.66% | | - | - | - | - | - | 64.54% |
| QCFS | 74.32% | ResNet-34 | - | - | - | 35.06% | 59.35% | 69.37% |
| **Ours** | 74.22% | | 69.11% | 72.66% | 73.81% | 74.17% | 74.14% | 73.93% |

2009) and ImageNet (Deng et al., 2009) datasets. The network architectures selected for evaluation include VGG-16 (Simonyan & Zisserman, 2014), ResNet-18, ResNet-20 and ResNet-34 (He et al., 2016). For the setting of the hyperparameter $\rho$, we set $\rho = 4$ for CIFAR-10/100 and $\rho = 8$ for ImageNet if there are no special instructions. More details of the experimental settings are provided in the Appendix.

## 5.1 EFFECTIVENESS OF THE PROPOSED METHOD

To illustrate the effectiveness of our proposed initial membrane potential shifting operations, we compare bar charts of offset spike $\psi^l$ in each layer of SNNs before and after the shift. Fig. 3 illustrated the results of VGG-16 networks on the CIFAR-10 and CIFAR-100 datasets. It can be observed that the shifting operations significantly reduce offset spike, that is, the deviation between $\phi^l(T)$ and $a^l$, for each layer. For vanilla settings without the shift operation (denoted as "w/o shift" in Fig. 3), one can discover a magnification effect of spike count error from the 1st to the 11th layer. In contrast, the apparent magnification is alleviated with the proposed methods. From Fig. 3(b), we notice that the $\pm 2$ and $\pm 3$ offset spike in the VGG-16 model for CIFAR-100 have increased compared to those for CIFAR-10 (Fig. 3(a)). Using our method can clearly decrease these deviations and achieve a comparable error-free conversion.

## 5.2 COMPARISON WITH STATE-OF-THE-ART METHODS

We compare our methods with previous state-of-the-art ANN-SNN conversion works, including RMP (Han et al., 2020), SNM (Wang et al., 2022a), SNNC-AP (Li et al., 2021), OPI (Bu et al., 2022a), QCFS (Bu et al., 2022b), on CIFAR-10, CIFAR-100 and ImageNet datasets. Since we spend $\rho$ time-steps in the first stage to acquire relevant temporal information about membrane potential, we

Table 2: Comparison with other types of SNN training methods

| Dataset | Method | Type | Architecture | Accuracy | Time-step |
|---|---|---|---|---|---|
| CIFAR-100 | Dual-Phase | Hybrid Training | VGG-16 | 70.08% | 4 |
| | Diet-SNN | Hybrid Training | VGG-16 | 69.67% | 5 |
| | RecDis-SNN | BPTT | VGG-16 | 69.88% | 5 |
| | **Ours**($\rho = 4$) | ANN-SNN conversion | VGG-16 | 74.24% | 1 |
| ImageNet | HC-STDB | Hybrid Training | ResNet-34 | 61.48% | 250 |
| | DSR | Supervised learning | PreAct-ResNet-18 | 67.74% | 50 |
| | STBP-tdBN | BPTT | ResNet-34 | 63.72% | 6 |
| | PLIF | BPTT | ResNet-34 | 67.04% | 7 |
| | TET | BPTT | ResNet-34 | 64.79% | 6 |
| | **Ours**($\rho = 4$) | ANN-SNN conversion | ResNet-34 | 67.12% | 2 |

Table 3: The ratio and MSE after multiple iterations

| Dataset | Architecture | Baseline | | **Ours** | | **Ours** $\times 2$ | | **Ours** $\times 4$ | |
|---|---|---|---|---|---|---|---|---|---|
| | | Ratio | MSE | Ratio | MSE | Ratio | MSE | Ratio | MSE |
| CIFAR-10 | VGG-16 | 88.33% | 0.120 | 97.65% | 0.024 | 99.83% | 0.002 | 99.84% | 0.002 |
| | ResNet-20 | 62.38% | 0.512 | 82.56% | 0.179 | 99.73% | 0.003 | 99.76% | 0.002 |
| CIFAR-100 | VGG-16 | 82.90% | 0.192 | 98.42% | 0.016 | 99.86% | 0.001 | 99.87% | 0.001 |
| | ResNet-20 | 41.59% | 1.641 | 67.08% | 0.453 | 86.03% | 0.165 | 91.29% | 0.101 |

will compare the performance of other works at time-step $T + \rho$ with our performance at time-step $T$ to ensure the fairness of comparison.

Tab. 1 reports the results on the CIFAR-100 dataset. For VGG-16, our method at time-step 1 ($\rho = 4$) outperforms SNM and SNNC-AP at time-step 32. Moreover, we achieve 76.26% top-1 accuracy with 4 time-steps ($\rho = 4$), which is 2.30% higher than QCFS (73.96%, T=8) and 15.77% higher than OPI (60.49%, T=8). For ResNet-20, the performance of our method at time-step 1 ($\rho = 4$) surpasses the performance of RMP at time-step 32 (59.22% vs. 27.64%). The accuracy of our method is 65.18% at time-step 4 ($\rho = 4$), whereas accuracies of QCFS and OPI are 55.37% and 23.09% at time-step 8, respectively. More results on CIFAR-10 are listed in the Appendix.

We further test the generalization of our method on the ImageNet (Tab. 1). For VGG-16, we achieve 73.82% top-1 accuracy at time-step 8 ($\rho = 8$), which outperforms QCFS (50.97%, T=16) by 22.85% and OPI (36.02%, T=16) by 37.80%. For ResNet-34, our method at time-step 1 ($\rho = 8$) outperforms SNM and SNNC-AP at time-step 32. Moreover, we achieve 74.17% with 8 time-steps ($\rho = 8$), which is 14.82% higher than QCFS (59.35%, T=16). These results show that our method can achieve better classification accuracy with fewer time-steps.

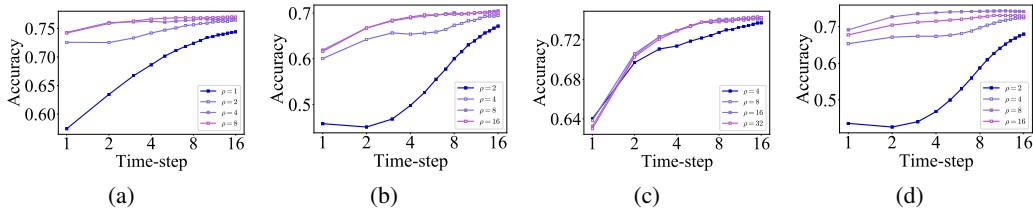

Figure 4: Influence of different $\rho$. (a) VGG-16 on CIFAR-100, (b) ResNet-20 on CIFAR-100, (c) VGG-16 on ImageNet, (d) ResNet-34 on ImageNet.

In addition, we compare our method with other types of SNN training methods (Hybrid Training & BPTT), including Dual-Phase (Wang et al., 2022b), Diet-SNN (Rathi & Roy, 2021), RecDis-SNN (Guo et al., 2022), HC-STDB (Rathi et al., 2020), STBP-tdBN (Zheng et al., 2021), PLIF (Fang et al., 2021), TET (Deng et al., 2022) and DSR (Meng et al., 2022a). Here we set $\rho = 4$ for the CIFAR-100 and ImageNet datasets. As reported in Tab. 2, our method achieves better accuracy on the CIFAR-100 dataset and comparable accuracy on the ImageNet dataset with the same quantity of time-steps. Note that compared to ANN-SNN conversion, the back-propagation approaches need to propagate the gradient through both spatial and temporal domains during the training process,

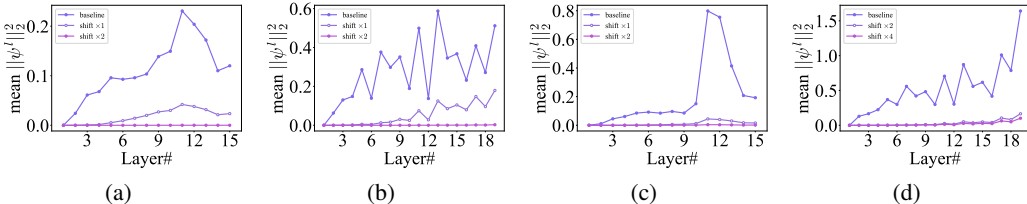

Figure 5: The MSE of conversion error after using iterative optimization. (a): VGG-16 on CIFAR-10, (b): ResNet-20 on CIFAR-10, (c): VGG-16 on CIFAR-100, (d): ResNet-20 on CIFAR-100.

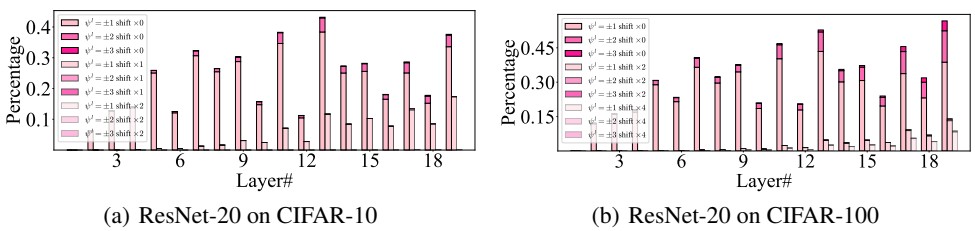

Figure 6: The distribution of offset spike after using iterative optimization

which consumes large amounts of memory and computing resources. All these results demonstrate the superiority of our method.

## 5.3 EFFECT OF THE INFERENCE TIME-STEP $\rho$

We further explore the influence of the hyperparameter $\rho$ in the first stage of our method. Fig. 4 shows the accuracy of the network with different values of $\rho$. For Fig.4(a)-4(d), the value of quantization level $L$ in QCFS function is set to 4, 8, 16 and 8. We find that the SNN accuracy tends to converge as $\rho$ gradually approaches $L$. This phenomenon can be understood as follows. We use the QCFS activation function in the source ANN and we have $\boldsymbol{a}^l \in \{k\theta^l/L|k = 0, 1, ..., L\}$ and $\boldsymbol{\phi}^l(T) \in \{k\theta^l/\rho|k = 0, 1, ..., \rho\}$. Thus, the mapping relationship between $\boldsymbol{a}^l$ and $\boldsymbol{\phi}^l(T)$ will become more accurate when $\rho$ approaches $L$, which makes the temporal information obtained from the first stage more precise to improve the performance of the network.

## 5.4 EFFECT OF THE ITERATIVE OPTIMIZATION

In section 4.4, we explain that our method has an iterative property that can reduce the offset spike through multiple iterations. To demonstrate this, we define the ratio as the percentage of $a_i^l = \phi_i^l(T)$ in the output layer. Despite this, we also consider the indicators of mean-square error (MSE), which is defined as $||\boldsymbol{\psi}^l||_2^2$. Tab. 3 reports the ratio and MSE of the output layer, in which the baseline denotes the performance without using our methods and $\times 2$ represents two iterations. Besides, we set $\rho = L = T$. From top to bottom in Tab. 3, the values of $L$ are set to 4, 4, 4 and 8. From Tab. 3 and Fig. 5, we can conclude that, generally, the ratio and MSE in each layer continue to decrease as the number of iterations increases, which is consistent with the results shown in Fig. 6.

## 6 CONCLUSIONS

In this paper, we first define offset spike to measure the degree of deviation between the actual and desired SNN firing rates. Then we analyse the distribution of offset spike and demonstrate that we can infer the specific value of the deviation according to the corresponding residual membrane potential. Furthermore, we propose an optimization method to eliminate offset spike by shifting the initial membrane potential up and down. Finally, we demonstrate the superiority of our method on CIFAR-10/100 and ImageNet datasets. Our results will further facilitate the relevant research and application of SNNs to neuromorphic chips.

ACKNOWLEDGMENTS

This work was supported by the National Natural Science Foundation of China under Grant No. 62176003 and No. 62088102.

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

## A  APPENDIX

### A.1  THE NETWORK CONFIGURATION IN THE TRAINING PROCEDURE

We choose Stochastic Gradient Descent optimizer (Bottou, 2012) and Cosine Annealing scheduler (Loshchilov & Hutter, 2017) to train ANN models for 300 epochs. For CIFAR-10/100, the value of weight decay is set to $5 \times 10^{-4}$, and the initial learning rates are 0.1 and 0.02, respectively. For ImageNet, we set the initial learning rate as 0.1 and weight decay as $1 \times 10^{-4}$. In addition, we adopt data-augmentation techniques (DeVries & Taylor, 2017; Cubuk et al., 2019; Li et al., 2021) to further improve the performance of the models.

In Fig. 1, we train the source ANN with the QCFS activation function (equation 7) and then convert it to an SNN. We set $T = L = 4$. For Figure 1(c)-1(d), we add the constraint that the output $a^{l-1}$ in layer $l-1$ of ANNs is the same as the output $\phi^{l-1}(T)$ of SNNs, that is, $a^{l-1} = \phi^{l-1}(T)$, and compute the offset spike with equation 8 and $a^l = f(W^l \phi^{l-1}(T))$.

### A.2  PROOF OF THEOREM

**Theorem 1.** *Supposing that an ANN with QCFS activation function (equation 7) is converted to an SNN with $L = T, \lambda^l = \theta^l, v^l(0) = \theta^l/2$, and the inputs to the l-th layer of ANN and SNN are the same, that is, $a^{l-1} = \phi^{l-1}(T)$. Then for any i-th element of the l-th layer, we will have the following conclusions:*
*(i) If $\phi_i^l(T) > 0$ and $v_i^l(T) < 0$, we will have $\phi_i^l(T) > a_i^l$ and $\psi_i^l < 0$.*
*(ii) If $\phi_i^l(T) < \theta^l$ and $v_i^l(T) \geqslant \theta^l$, we will have $\phi_i^l(T) < a_i^l$ and $\psi_i^l > 0$.*

*Proof.* According to the preconditions and equation 5, we have:

$$\phi_i^l(T) = \frac{\sum\limits_{t=1}^{T} I_i^l(t)}{T} - \frac{v_i^l(T) - \theta^l/2}{T}. \tag{S1}$$

If $\sum\limits_{t=1}^{T} I_i^l(t) \in [-\theta^l/2, \theta^l T + \theta^l/2)$, based on the preconditions and equation 7, we get:

$$a_i^l = \frac{\theta^l}{T} \left\lfloor \frac{\sum\limits_{t=1}^{T} I_i^l(t)}{\theta^l} + \frac{1}{2} \right\rfloor. \tag{S2}$$

When $\sum\limits_{t=1}^{T} I_i^l(t) \in [k\theta^l - \theta^l/2, k\theta^l + \theta^l/2), k = 0, 1, ..., T$, from equation S2 we will have $a_i^l = k\theta^l/T$. For (i), by combining $v_i^l(T) < 0$ and equation S1 we will have:

$$\begin{aligned} \phi_i^l(T) &= \frac{\sum\limits_{t=1}^{T} I_i^l(t)}{T} - \frac{v_i^l(T) - \theta^l/2}{T} \\ &> \sum\limits_{t=1}^{T} I_i^l(t)/T + \theta^l/2T \\ &\geqslant k\theta^l/T = a_i^l. \end{aligned} \tag{S3}$$

When $\sum\limits_{t=1}^{T} I_i^l(t) < -\theta^l/2, a_i^l = 0$, according to the precondition $\phi_i^l(T) > 0$, we have $\phi_i^l(T) > a_i^l$.

In addition, if $\sum\limits_{t=1}^{T} I_i^l(t) \geqslant \theta^l T + \theta^l/2$, according to $v_i^l(T) < 0$ and equation S1, $\phi_i^l(T) > \theta^l$, which is impossible. Therefore, we can derive that $\phi_i^l(T) > a_i^l, \psi_i^l = (a_i^l - \phi_i^l(T))T/\theta^l < 0$ and we have already proved (i).

For (ii), we also first consider $\sum_{t=1}^{T} I_i^l(t) \in [-\theta^l/2, \theta^l T + \theta^l/2)$. When $\sum_{t=1}^{T} I_i^l(t) \in [k\theta^l - \theta^l/2, k\theta^l + \theta^l/2)$, $k = 0, 1, ..., T$, from equation S2 we will have $a_i^l = k\theta^l/T$, by combining $v_i^l(T) \geqslant \theta^l$ and equation S1, we will have:

$$
\begin{aligned}
\phi_i^l(T) &= \frac{\sum_{t=1}^{T} I_i^l(t)}{T} - \frac{v_i^l(T) - \theta^l/2}{T} \\
&\leqslant \sum_{t=1}^{T} I_i^l(t)/T - \theta^l/2T. \\
&< k\theta^l/T = a_i^l
\end{aligned}
\tag{S4}
$$

When $\sum_{t=1}^{T} I_i^l(t) \geqslant \theta^l T + \theta^l/2$, $a_i^l = \theta^l$, according to the precondition $\phi_i^l(T) < \theta^l$, we have $\phi_i^l(T) < a_i^l$. In addition, if $\sum_{t=1}^{T} I_i^l(t) < -\theta^l/2$, according to $v_i^l(T) \geqslant \theta^l$ and equation S1, $\phi_i^l(T) < 0$, which is impossible. Therefore, we can derive that $\phi_i^l(T) < a_i^l$, $\psi_i^l = (a_i^l - \phi_i^l(T))T/\theta^l > 0$ and we have proved (ii). □

**Theorem 2.** *If we use $s_i^l(t)$ and $\widetilde{s}_i^l(t)$ to denote the $i$-th element in the binary output of the $l$-th layer at time-step $t$ before and after optimization, $v_i^l(0)$ and $\widetilde{v}_i^l(0)$ to represent the initial membrane potential before and after optimization, then $\forall \epsilon \in (0, \theta^l)$, we will have the following conclusions:*

*(i) If we set $\widetilde{v}_i^l(0) = v_i^l(0) - \max\left(\theta^l, \min\{v_i^l(t)|s_i^l(t) = 1\} + \epsilon\right)$, then $\sum_{t=1}^{T} \widetilde{s}_i^l(t) = \sum_{t=1}^{T} s_i^l(t) - 1$.*

*(ii) If we set $\widetilde{v}_i^l(0) = v_i^l(0) + \max\left(\theta^l, \theta^l + \epsilon - \max\{v_i^l(t)|s_i^l(t) = 0\}\right)$, then $\sum_{t=1}^{T} \widetilde{s}_i^l(t) = \sum_{t=1}^{T} s_i^l(t) + 1$.*

We use $m_i^l(t), \widetilde{m}_i^l(t)$ to represent the accumulative potential at time-step $t$ before and after using optimization. Before the proof of theorem, we firstly introduce Lemma 1.

**Lemma 1.** *For situation (i) in Theorem 2, $\exists t \in [1, T], \sum_{k=1}^{t} s_i^l(k) = \sum_{k=1}^{t} \widetilde{s}_i^l(k) + 1$. For situation (ii) in Theorem 2, $\exists t \in [1, T], \sum_{k=1}^{t} s_i^l(k) = \sum_{k=1}^{t} \widetilde{s}_i^l(k) - 1$.*

*Proof.* For situation (i) in Theorem 2, we use $t_o$ to denote the specific time when $v_i^l(t_o) = \min\{v_i^l(t)|s_i^l(t) = 1\} \wedge s_i^l(t_o) = 1$. $\forall t$, as we optimize SNNs layer by layer, we have the following equation:

$$
m_i^l(t) = v_i^l(0) + \sum_{k=1}^{t} I_i^l(k) - \sum_{k=1}^{t-1} s_i^l(k)\theta^l,
\tag{S5}
$$

$$
\widetilde{m}_i^l(t) = \widetilde{v}_i^l(0) + \sum_{k=1}^{t} I_i^l(k) - \sum_{k=1}^{t-1} \widetilde{s}_i^l(k)\theta^l.
\tag{S6}
$$

As $v_i^l(0) > \widetilde{v}_i^l(0)$ and we use the same input $\sum_{k=1}^{t} I_i^l(k)$ before and after optimization, when $\sum_{k=1}^{t-1} s_i^l(k)\theta^l = \sum_{k=1}^{t-1} \widetilde{s}_i^l(k)\theta^l$, we will have $m_i^l(t) > \widetilde{m}_i^l(t)$ and further derive $s_i^l(t) \geqslant \widetilde{s}_i^l(t)$, which means that $\forall t, \sum_{k=1}^{t} s_i^l(k)\theta^l \geqslant \sum_{k=1}^{t} \widetilde{s}_i^l(k)\theta^l$.

If $\exists t' \in [1, t_o), \sum_{k=1}^{t'} s_i^l(k) = \sum_{k=1}^{t'} \widetilde{s}_i^l(k) + 1$, then we have already found a qualified time $t'$. If $\sum_{k=1}^{t_o-1} s_i^l(k) = \sum_{k=1}^{t_o-1} \widetilde{s}_i^l(k)$, we will have:

$$
\begin{aligned}
m_i^l(t_o) - \widetilde{m}_i^l(t_o) &= v_i^l(0) - \widetilde{v}_i^l(0) \\
&= \max\left(\theta^l, \min\left\{v_i^l(t)|s_i^l(t) = 1\right\} + \epsilon\right) \\
&= \max\left(\theta^l, v_i^l(t_o) + \epsilon\right).
\end{aligned}
\tag{S7}
$$

As $m_i^l(t_o) = v_i^l(t_o) + \theta^l$, we will further have:

$$
\begin{aligned}
\widetilde{m}_i^l(t_o) &= m_i^l(t_o) - \max\left(\theta^l, v_i^l(t_o) + \epsilon\right) \\
&\leqslant m_i^l(t_o) - v_i^l(t_o) - \epsilon \\
&< \theta^l.
\end{aligned}
\tag{S8}
$$

From the above equation, we can derive $\widetilde{m}_i^l(t_o) < \theta^l$ and $s_i^l(t_o) = 1, \widetilde{s}_i^l(t_o) = 0$, then we will have $\sum_{k=1}^{t_o} s_i^l(k) = \sum_{k=1}^{t_o} \widetilde{s}_i^l(k) + 1$, which means that $t_o$ is a qualified time.

For situation (ii) in Theorem 2, we use $t_o$ to denote the specific time when $v_i^l(t_o) = \max\left\{v_i^l(t)|s_i^l(t) = 0\right\} \wedge s_i^l(t_o) = 0$. Similarly, we will derive $\forall t, \sum_{k=1}^{t} s_i^l(k)\theta^l \leqslant \sum_{k=1}^{t} \widetilde{s}_i^l(k)\theta^l$ according to equation S5-equation S6.

If $\exists t' \in [1, t_o), \sum_{k=1}^{t'} s_i^l(k) = \sum_{k=1}^{t'} \widetilde{s}_i^l(k) - 1$, then we have already found a qualified time $t'$. If $\sum_{k=1}^{t_o-1} s_i^l(k) = \sum_{k=1}^{t_o-1} \widetilde{s}_i^l(k)$, we will have:

$$
\begin{aligned}
\widetilde{m}_i^l(t_o) - m_i^l(t_o) &= \widetilde{v}_i^l(0) - v_i^l(0) \\
&= \max\left(\theta^l, \theta^l + \epsilon - \max\left\{v_i^l(t)|s_i^l(t) = 0\right\}\right) \\
&= \max\left(\theta^l, \theta^l + \epsilon - v_i^l(t_o)\right).
\end{aligned}
\tag{S9}
$$

As $m_i^l(t_o) = v_i^l(t_o)$, we will further have:

$$
\begin{aligned}
\widetilde{m}_i^l(t_o) &= m_i^l(t_o) + \max\left(\theta^l, \theta^l + \epsilon - v_i^l(t_o)\right) \\
&\geqslant m_i^l(t_o) + \theta^l + \epsilon - v_i^l(t_o) \\
&> \theta^l.
\end{aligned}
\tag{S10}
$$

From the above equation, we can derive $\widetilde{m}_i^l(t_o) > \theta^l$ and $s_i^l(t_o) = 0, \widetilde{s}_i^l(t_o) = 1$, then we will have $\sum_{k=1}^{t_o} s_i^l(k) = \sum_{k=1}^{t_o} \widetilde{s}_i^l(k) - 1$, which means that $t_o$ is a qualified time. $\qquad \square$

Now we will further prove Theorem 2.

*Proof.* **Proof of (i).** If $\theta^l > \min\left\{v_i^l(t)|s_i^l(t) = 1\right\} + \epsilon, \widetilde{v}_i^l(0) = v_i^l(0) - \theta^l$. According to Lemma 1, $\exists t_s, \sum_{k=1}^{t_s} s_i^l(k) = \sum_{k=1}^{t_s} \widetilde{s}_i^l(k) + 1$, then we will have $m_i^l(t_s+1) = \widetilde{m}_i^l(t_s+1)$ by combining equation S5 and equation S6, which means that $\sum_{k=t_s+1}^{T} s_i^l(k) = \sum_{k=t_s+1}^{T} \widetilde{s}_i^l(k)$ in the remaining time cycle. As a result, we can have $\sum_{k=1}^{T} s_i^l(k) = \sum_{k=1}^{T} \widetilde{s}_i^l(k) + 1$.

If $\theta^l < \min \{v_i^l(t)|s_i^l(t) = 1\} + \epsilon$, $\widetilde{v}_i^l(0) = v_i^l(0) - \min \{v_i^l(t)|s_i^l(t) = 1\} - \epsilon$. According to Lemma 1, $\exists t_s, \sum\limits_{k=1}^{t_s} s_i^l(k) = \sum\limits_{k=1}^{t_s} \widetilde{s}_i^l(k) + 1$, then we will have $m_i^l(t_s + 1) = \widetilde{m}_i^l(t_s + 1) + \min \{v_i^l(t)|s_i^l(t) = 1\} + \epsilon - \theta^l$, which means that $m_i^l(t_s + 1) > \widetilde{m}_i^l(t_s + 1)$.

For $m_i^l$, if we set $t'$ as the first spike firing time after $t_s$, which means that $m_i^l(t') = v_i^l(t') + \theta^l$ and $\sum\limits_{k=1}^{t'-1} s_i^l(k) = \sum\limits_{k=1}^{t'-1} \widetilde{s}_i^l(k) + 1$, then we will have $\widetilde{m}_i^l(t') = m_i^l(t') - \min \{v_i^l(t)|s_i^l(t) = 1\} - \epsilon + \theta^l = v_i^l(t') - \min \{v_i^l(t)|s_i^l(t) = 1\} - \epsilon + 2\theta^l > \theta^l$. which means that $s_i^l(t') = \widetilde{s}_i^l(t') = 1$, $\sum\limits_{k=1}^{t'} s_i^l(k) = \sum\limits_{k=1}^{t'} \widetilde{s}_i^l(k) + 1$. If we continue to use the above derivation process, we can finally have $\sum\limits_{k=1}^{T} s_i^l(k) = \sum\limits_{k=1}^{T} \widetilde{s}_i^l(k) + 1$.

**Proof of (ii).** If $\theta^l > \theta^l + \epsilon - \max \{v_i^l(t)|s_i^l(t) = 0\}$, $\widetilde{v}_i^l(0) = v_i^l(0) + \theta^l$. According to Lemma 1, $\exists t_s, \sum\limits_{k=1}^{t_s} s_i^l(k) = \sum\limits_{k=1}^{t_s} \widetilde{s}_i^l(k) - 1$, then we will have $m_i^l(t_s + 1) = \widetilde{m}_i^l(t_s + 1)$ by combining equation S5 and equation S6, which means that $\sum\limits_{k=t_s+1}^{T} s_i^l(k) = \sum\limits_{k=t_s+1}^{T} \widetilde{s}_i^l(k)$ in the remaining time cycle. As a result, we can have $\sum\limits_{k=1}^{T} s_i^l(k) = \sum\limits_{k=1}^{T} \widetilde{s}_i^l(k) - 1$.

If $\theta^l < \theta^l + \epsilon - \max \{v_i^l(t)|s_i^l(t) = 0\}$, $\widetilde{v}_i^l(0) = v_i^l(0) + \theta^l + \epsilon - \max \{v_i^l(t)|s_i^l(t) = 0\}$. According to Lemma 1, $\exists t_s, \sum\limits_{k=1}^{t_s} s_i^l(k) = \sum\limits_{k=1}^{t_s} \widetilde{s}_i^l(k) - 1$, then we will have $m_i^l(t_s + 1) = \widetilde{m}_i^l(t_s + 1) - \epsilon + \max \{v_i^l(t)|s_i^l(t) = 0\}$, which means that $m_i^l(t_s + 1) < \widetilde{m}_i^l(t_s + 1)$.

For $\widetilde{m}_i^l$, if we set $t'$ as the first spike firing time after $t_s$, which means that $\widetilde{m}_i^l(t') = \widetilde{v}_i^l(t') + \theta^l$ and $\sum\limits_{k=1}^{t'-1} s_i^l(k) = \sum\limits_{k=1}^{t'-1} \widetilde{s}_i^l(k) - 1$. Similar to situation (i), we need to prove that $s_i^l(t') = \widetilde{s}_i^l(t') = 1$. However, it is not easy to make a direct proof. Therefore, we attempt to prove its inverse and negative thesis : when $t' > t_s \wedge \sum\limits_{k=1}^{t'-1} s_i^l(k) = \sum\limits_{k=1}^{t'-1} \widetilde{s}_i^l(k) - 1$, if $s_i^l(t') = 0$, then we can have $\widetilde{s}_i^l(t') = 0$.

Under this condition, we can derive $m_i^l(t') = v_i^l(t') \wedge m_i^l(t') = \widetilde{m}_i^l(t') - \epsilon + \max \{v_i^l(t)|s_i^l(t) = 0\}$, then we will have $\widetilde{m}_i^l(t') = v_i^l(t') + \epsilon - \max \{v_i^l(t)|s_i^l(t) = 0\}$. As $\max \{v_i^l(t)|s_i^l(t) = 0\} \geqslant v_i^l(t')$, $\widetilde{m}_i^l(t') \leqslant \epsilon < \theta^l$, which means that $\widetilde{s}_i^l(t') = 0$.

Therefore, if we set $t'$ as the first spike firing time for $\widetilde{m}_i^l$ after $t_s$, we can prove that $s_i^l(t') = \widetilde{s}_i^l(t') = 1$, which means that $\sum\limits_{k=1}^{t'} s_i^l(k) = \sum\limits_{k=1}^{t'} \widetilde{s}_i^l(k) - 1$. If we continue to use the above derivation process, we can finally have $\sum\limits_{k=1}^{T} s_i^l(k) = \sum\limits_{k=1}^{T} \widetilde{s}_i^l(k) - 1$. □

## A.3 EXPERIMENTAL RESULTS ON CIFAR-10 DATASET

Tab. S1 reports the results on CIFAR-10 dataset. For VGG-16, the accuracy of our proposed method is 95.46% with 4 time-step ($\rho = 4$), whereas the accuracies of OPI and QCFS are 90.96% and 94.95% with 8 time-step, respectively. For ResNet-18, we achieve 95.46% with 4 time-steps ($\rho = 4$), whereas the corresponding performance of OPI and QCFS are 75.44% and 95.04%. For ResNet-20, our method reaches 91.68% with 4 time-steps ($\rho = 4$), which is 2.13% higher than QCFS (89.55%, T=8) and 25.44% higher than OPI (66.24%, T=8).

Table S1: Comparison with other ANN-SNN conversion methods on CIFAR-10 dataset

| Method | ANN | Architecture | T=1 | T=2 | T=4 | T=8 | T=16 | T=32 |
|---|---|---|---|---|---|---|---|---|
| SNM | 94.09% | | - | - | - | - | - | 93.43% |
| SNNC-AP | 95.72% | | - | - | - | - | - | 93.71% |
| OPI | 94.57% | VGG-16 | - | - | - | 90.96% | 93.38% | 94.20% |
| QCFS | 95.52% | | - | 91.18% | 93.96% | 94.95% | 95.40% | 95.54% |
| **Ours** | 95.51% | | 94.90% | 95.36% | 95.46% | 95.51% | 95.57% | 95.61% |
| SNM | 95.39% | | - | - | - | - | - | 94.03% |
| SNNC-AP | 95.46% | | - | - | - | - | - | 94.78% |
| OPI | 96.04% | ResNet-18 | - | - | - | 75.44% | 90.43% | 94.82% |
| QCFS | 95.64% | | - | 91.75% | 93.83% | 95.04% | 95.56% | 95.67% |
| **Ours** | 95.64% | | 95.25% | 95.45% | 95.46% | 95.66% | 95.68% | 95.68% |
| OPI | 92.74% | | - | - | - | 66.24% | 87.22% | 91.88% |
| QCFS | 91.77% | ResNet-20 | - | 73.20% | 83.75% | 89.55% | 91.62% | 92.24% |
| **Ours** | 91.77% | | 89.88% | 91.26% | 91.68% | 91.86% | 92.20% | 92.16% |

### A.4 ELIMINATING OFFSET SPIKE THROUGH ITERATIVE OPTIMIZATION

In Sections 4.4 and 5.4, we have pointed out the iterative property of our proposed method. Here we will make a discussion in detail. Firstly, we can infer the specific value of $\psi^l$ based on the residual membrane potential when the corresponding input current belongs to a specific interval, which is illustrated in the following theorem.

**Theorem 3.** *Supposing that an ANN with QCFS activation function (equation 7) is converted to an SNN with $L = T, \lambda^l = \theta^l, v^l(0) = \theta^l/2$, and the inputs to the $l$-th layer of ANN and SNN are the same, that is, $a^{l-1} = \phi^{l-1}(T)$. Then for any $i$-th element of the $l$-th layer, we will have the following conclusions:*

*If $\sum_{t=1}^{T} I_i^l(t) \in [-\theta^l/2, \theta^l T + \theta^l/2)$, when $v_i^l(T)/\theta^l \in [k, k+1)$, we will have $\psi_i^l = a_i^l T/\theta^l - \sum_{t=1}^{T} s_i^l(t) = k$, where $k \in \mathbb{Z}$.*

*Proof.* As the preconditions of Theorem 3 are same as the preconditions of equation S1 and equation S2, by combining equation S1 and equation S2, we will have:

$$a_i^l T/\theta^l - \sum_{t=1}^{T} s_i^l(t) = \left\lfloor \frac{\sum_{t=1}^{T} I_i^l(t)}{\theta^l} + \frac{1}{2} \right\rfloor - \frac{T}{\theta^l} \left( \frac{\sum_{t=1}^{T} I_i^l(t)}{T} - \frac{v_i^l(T) - \theta^l/2}{T} \right)$$

$$= v_i^l(T)/\theta^l + \left\lfloor \sum_{t=1}^{T} I_i^l(t)/\theta^l + 1/2 \right\rfloor - \left( \sum_{t=1}^{T} I_i^l(t)/\theta^l + 1/2 \right). \quad \text{(S11)}$$

As $-1 < \left\lfloor \sum_{t=1}^{T} I_i^l(t)/\theta^l + 1/2 \right\rfloor - \left( \sum_{t=1}^{T} I_i^l(t)/\theta^l + 1/2 \right) \leqslant 0$, when $v_i^l(T)/\theta^l \in [k, k+1), k-1 < a_i^l T/\theta^l - \sum_{t=1}^{T} s_i^l(t) < k+1$. Considering that $a_i^l T/\theta^l - \sum_{t=1}^{T} s_i^l(t) \in \mathbb{Z}$, we have $\psi_i^l = a_i^l T/\theta^l - \sum_{t=1}^{T} s_i^l(t) = k$. $\square$

In fact, even if the input current does not belong to the specific interval, from equation 7, we can derive that when $\sum_{t=1}^{T} \boldsymbol{I}^l(t) < -\theta^l/2, \boldsymbol{a}^l = 0$ and when $\sum_{t=1}^{T} \boldsymbol{I}^l(t) \geqslant \theta^l T + \theta^l/2, \boldsymbol{a}^l = \theta^l$, then we

Table S2: Input/Output Ratio for each layer of an SNN with VGG-16 on CIFAR-10 dataset

| Condition | L1 | L2 | L3 | L4 | L5 | L6 | L7 | L8 | L9 | L10 | L11 | L12 | L13 | L14 | L15 |
|---|---|---|---|---|---|---|---|---|---|---|---|---|---|---|---|
| Input Ratio | 100% | 99.99% | 99.97% | 99.90% | 99.63% | 99.24% | 98.77% | 97.84% | 97.43% | 97.28% | 97.31% | 97.52% | 97.11% | 97.88% | 97.80% |
| Output Ratio | 100% | 99.99% | 99.99% | 99.99% | 99.99% | 99.99% | 99.98% | 99.95% | 99.94% | 99.90% | 99.68% | 99.68% | 99.74% | 99.85% | 99.87% |
| Output Ratio when Input is accurate | 100% | 100% | 100% | 100% | 100% | 100% | 100% | 100% | 100% | 100% | 100% | 100% | 100% | 100% | 100% |

Table S3: Comparison with different initialization strategies

| Method | Dataset | Method | T=1 | T=2 | T=4 | T=8 | T=16 |
|---|---|---|---|---|---|---|---|
| CIFAR-100 | VGG-16 | Random Intialization | - | - | - | 68.03% | 74.74% |
| CIFAR-100 | VGG-16 | $v^l(0) \leftarrow \theta^l/2$ | - | 63.79% | 69.62% | 73.96% | 76.24% |
| CIFAR-100 | VGG-16 | $v^l(0) \leftarrow v^l(\rho), \rho = 4$ | 73.38% | 74.53% | 75.09% | 76.27% | 76.62% |
| CIFAR-100 | VGG-16 | **Ours**$(\rho = 4)$ | 74.24% | 76.03% | 76.26% | 76.52% | 76.77% |

can also directly determine the $\psi^l$ according to the value of $\phi^l(T)$. After we have already acquired the value of $\psi^l$, we will adopt our optimization method for $|\psi_i^l|$ times to eliminate the offset spike on $i$-th element neuron of the $l$-th layer.

In Tab. 3, the Ratio after multiple iterations does not achieve 100%. We find that the non-zero MSE and Ratio in Tab. 3 are caused by the rounding of the floating-point numbers. Specifically, we carefully checked the Ratio, defined as the percentage of SNN input (output) equals ANN input (output) in each layer, to prove this, and we list the results in Tab. S2.

We find that the Ratio of the output in layer 1 is 100%, but the Ratio of the input in layer 2 is close to 100%. Thus, the error must be caused by the floating point number precision problem in multiplication and division operations involved in the forward propagation between layer 1 and layer 2. Considering that SNNs will calculate $\sum_{t=1}^{T} \boldsymbol{W}^l \boldsymbol{s}^{l-1}(t)/T$ but ANNs will calculate $\boldsymbol{W}^l(\sum_{t=1}^{T} \boldsymbol{s}^{l-1}(t)/T)$ as the average input current for the $l$-th layer, these two corresponding inputs are not necessarily equal due to the rounding of the floating point number.

We then conduct another experiment to prove that conversion errors can be reduced to zero if the rounding of the floating point number is eliminated. We force the input of spiking neurons to be the same as QCFS neurons in each layer and calculate the Ratio of the output. As shown in Tab. S2 (line 4), we find that the Ratio of the output in each SNN layer is 100%, which indicates that iterating the proposed method can finally reduce conversion error to zero.

## A.5 COMPARISON WITH DIFFERENT INITIALIZATION STRATEGIES

We make a comparison among different initialization strategies on CIFAR-100 with VGG-16 structure, including random initialization, setting $v^l(0) = \theta^l/2$ (Bu et al., 2022b), using the residual membrane potential $v^l(\rho)$ of the first stage as the initial membrane potential and our proposed method. As shown in Table S3, our proposed method outperforms other initialization strategies under low time-steps, which proves the superiority of our method.

From Tab. S3 (line 4), we notice that using the residual membrane potential $v^l(\rho)$ as the initial membrane potential also achieves considerable performance. Therefore, besides our proposed method, we can also provide a lightweight optimization scheme: for each layer, we can consider directly selecting the residual membrane potential $\boldsymbol{v}^l(\rho)$ after $\rho$ steps as the initial membrane potential for our second stage. The idea is to make $\boldsymbol{v}^l(T) - \boldsymbol{v}^l(0)$ in equation 5 approach 0 to eliminate conversion errors (offset spike). Tab. S4 reports further results on the ImageNet dataset. Although the performance of our lightweight optimization scheme is weaker than our best solution, it is still much better than the current SOTA methods. Under the condition of using the lightweight scheme, we can avoid the extra calculation of the optimal shifting distance. We compare the running time among QCFS, our lightweight scheme, and our shifting method on CIFAR-100 with VGG-16 structure and 16 time-steps. The corresponding running time is 101s, 101s, and 134s, respectively.

Table S4: Comparison with the state-of-the-art ANN-SNN conversion methods

| Dataset | Architecture | Method | T=1 | T=2 | T=4 | T=8 | T=16 |
|---|---|---|---|---|---|---|---|
| ImageNet | VGG-16 | OPI | - | - | - | 6.25% | 36.02% |
| | | QCFS | - | - | - | 19.12% | 50.97% |
| | | lightweight(ours) | 62.27% | 69.69% | 72.50% | 73.39% | 74.04% |
| | | shifting(ours) | 63.84% | 70.59% | 72.94% | 73.82% | 74.09% |
| | ResNet-34 | QCFS | - | - | - | 35.06% | 59.35% |
| | | lightweight(ours) | 69.04% | 69.63% | 69.80% | 69.77% | 70.97% |
| | | shifting(ours) | 69.11% | 72.66% | 73.81% | 74.17% | 74.14% |

## A.6 PSEUDO-CODE FOR OVERALL ALGORITHM FLOW

---

**Algorithm 1** Algorithm for ANN-SNN conversion.

---

**Require:** The quantity of time-steps to calculate residual membrane potential $\rho$; The quantity of time-steps to test dataset $T$; The iteration number of the optimization strategy **ItNum**; The corresponding input for SNN layer $l$ **data**$^l$; The shifting variable mentioned in Theorem 2 $\epsilon$; Pretrained QCFS ANN model $f_{\text{ANN}}(\boldsymbol{W}, \lambda)$; Dataset $D$.

**Ensure:** SNN model $f_{\text{SNN}}(\boldsymbol{W}, \theta, \boldsymbol{v}, \boldsymbol{s})$.

1: # Convert ANN to SNN
2: **for** $l = 1$ to $f_{\text{ANN}}$.layers **do**
3:      $f_{\text{SNN}}.\theta^l = f_{\text{ANN}}.\lambda^l$
4:      $f_{\text{SNN}}.\boldsymbol{v}^l(0) = \frac{1}{2} f_{\text{SNN}}.\theta^l$
5:      $f_{\text{SNN}}.\boldsymbol{W}^l = f_{\text{ANN}}.\boldsymbol{W}^l$
6: **end for**
7: # Eliminate offset spike
8: **for** (**Image**, **label**) in $D$ **do**
9:      **for** $t = 1$ to $T$ **do**
10:          **data**$^1(t) = $ **Image**
11:      **end for**
12:      **for** $l = 1$ to $f_{\text{SNN}}$.layers **do**
13:          **for** $epoch = 1$ to **ItNum** **do**
14:              # Acquire the residual membrane potential
15:              **for** $t = 1$ to $\rho$ **do**
16:                  $f_{\text{SNN}}.\boldsymbol{s}^l((epoch - 1) \times \rho + t) = f_{\text{SNN}}^l(\textbf{data}^l(t))$
17:              **end for**
18:              # Optimize the initial membrane potential with the optimal shifting distance
19:              **if** Need to shift the initial membrane potential up according to Theorem 3 **then**
20:                  $f_{\text{SNN}}.\boldsymbol{v}^l(epoch \times \rho) = f_{\text{SNN}}.\boldsymbol{v}^l((epoch - 1) \times \rho) + \max(\theta^l, \theta^l + \epsilon - \max\{f_{\text{SNN}}.\boldsymbol{v}^l(t) | f_{\text{SNN}}.\boldsymbol{s}^l(t) = 0, t \in [(epoch - 1) \times \rho + 1, epoch \times \rho]\})$
21:              **end if**
22:              **if** Need to shift the initial membrane potential down according to Theorem 3 **then**
23:                  $f_{\text{SNN}}.\boldsymbol{v}^l(epoch \times \rho) = f_{\text{SNN}}.\boldsymbol{v}^l((epoch - 1) \times \rho) - \max(\theta^l, \epsilon + \min\{f_{\text{SNN}}.\boldsymbol{v}^l(t) | f_{\text{SNN}}.\boldsymbol{s}^l(t) = 1, t \in [(epoch - 1) \times \rho + 1, epoch \times \rho]\})$
24:              **end if**
25:          **end for**
26:          **for** $t = 1$ to $T$ **do**
27:              $f_{\text{SNN}}.\boldsymbol{s}^l(\textbf{ItNum} \times \rho + t) = f_{\text{SNN}}^l(\textbf{data}^l(t))$
28:              $\textbf{data}^{l+1}(t) = f_{\text{SNN}}.\boldsymbol{W}^l(f_{\text{SNN}}.\boldsymbol{s}^l(\textbf{ItNum} \times \rho + t) f_{\text{SNN}}.\theta^l)$
29:          **end for**
30:      **end for**
31: **end for**
32: **return** $f_{\text{SNN}}(\boldsymbol{W}, \theta, \boldsymbol{v}, \boldsymbol{s})$

---

