# OpenReview forum: "Bridging the Gap between ANNs and SNNs by Calibrating Offset Spikes"
_ICLR.cc/2023/Conference — ICLR 2023 poster_

### Official Review · Reviewer_EoJ9 · 2022-10-13

**Confidence:** 4
**Correctness:** 3
**Technical Novelty And Significance:** 3
**Empirical Novelty And Significance:** 3
**Recommendation:** 6

**Clarity, Quality, Novelty And Reproducibility:**

Clarity: 6/10

Quality: 6/10

Novelty: 6/10

Reproducibility: 6/10

**Strength And Weaknesses:**

Strengths:
1. The tackled problem is relevant to the scope of ICLR.
2. The results look promising.

Weaknesses:
1. It is recommended to change the title to avoid confusion with the following paper:
Yang Li, Xiang He, Yiting Dong, Qingqun Kong, Yi Zeng. Spike Calibration: Fast and Accurate Conversion of Spiking Neural Network for Object Detection and Segmentation. CoRR abs/2207.02702 (2022).
2. In Section 2: "Inspired by these works, we incorporate the concept of calibration spike time into the conversion pipeline to bridge the gap between ANNs and SNNs." This sentence is too vague. The differences between the related works and the proposed work should be properly discussed in detail.
3. All the figures are too small (in particular the legends). It is recommended to increase their size to make them readable.
4. The description of the proposed method in Section 4 looks too vague and hard to follow. It is recommended to use algorithms, schemes, examples to ease the discussion. (To save space, the content of Section 3 can be moved to the appendix.)
5. It would be useful to provide the source code for reviewers' inspection during the rebuttal.

**Summary Of The Paper:**

The paper proposes an efficient ANN-SNN conversion method based on Spike Calibration. The experiments show that the conversion using the proposed approach reaches competitive accuracy with few time-steps.

**Summary Of The Review:**

Borderline paper where several concerns should be clarified.

---

> ### Author Response · Authors · 2022-11-17
> **Response to Reviewer EoJ9**
>
> Thanks for your valuable and constructive feedback. We are delighted that you find our results look promising, and the proposed approach reaches competitive accuracy with few time-steps. We would like to address your concerns and answer your questions in the following.
>
> >1. It is recommended to change the title to avoid confusion with the following paper.
>
>
> To avoid confusion with the reference [1] you suggested, we decided to change the title to "Bridging the Gap between ANNs and SNNs by Calibrating Offset Spikes", and have revised it in the newly submitted version. As the OpenReview system currently does not support title change, we will change it in the final version.
>
> [1] Yang Li, Xiang He, Yiting Dong, Qingqun Kong, Yi Zeng. Spike calibration: Fast and accurate conversion of spiking neural network for object detection and segmentation. arXiv preprint arXiv:2207.02702 (2022).
>
>
> >2. In Section 2: "Inspired by these works, we incorporate the concept of calibration spike time into the conversion pipeline to bridge the gap between ANNs and SNNs." This sentence is too vague. The differences between the related works and the proposed work should be properly discussed in detail.
>
>
>
> Thanks for your constructive advice. We have revised this sentence and made a detailed explanation of the differences between related works and our proposed work in our newly submitted version. We would like to give an explanation of this sentence:
>
> The works on backpropagation with exact spike timing listed in the second paragraph of Related Works alter weights when training, and they all stress the importance of spike timing. Yet timing is usually ignored in conversion methods. Setting the initial membrane potential can change the spike timing. When the first spike time is delayed and the number of inference time-steps is limited, the firing rate may decrease. Inspired by this fact, we incorporate the concept of calibration spike time into the conversion pipeline to bridge the gap between ANNs and SNNs.
>
>
> >3. All the figures are too small (in particular the legends). It is recommended to increase their size to make them readable.
>
> Thanks for your valuable suggestion. We have modified all figures in the revised paper to make it readable.
>
> >4. The description of the proposed method in Section 4 looks too vague and hard to follow. It is recommended to use algorithms, schemes, examples to ease the discussion. (To save space, the content of Section 3 can be moved to the appendix.)
>
>
> Thanks for pointing it out! We have rewritten Section 4 according to your suggestion. In section 4.1, we added more explanation of Definition 1 and Observations 1-2 to make it easier to understand. In section 4.3, we used examples to demonstrate that our theorem gives the optimal shifting distance to move the offset spike by $\pm 1$. Besides, the details of the proposed ANN-SNN algorithm are added in the Appendix due to the limited space. We would like to clarify that we did not choose to move Section 3 to the Appendix, as many equations in Section 4 follow the definition in Section 3.
>
>
> >5. It would be useful to provide the source code for reviewers' inspection during the rebuttal.
>
> We would like to clarify that we have attached the code in supplementary files. Please refer to the zip file of Supplementary Material under the abstract.

---

> > ### Comment · Reviewer_EoJ9 · 2022-11-21
> > **Response to Authors**
> >
> > The efforts made by the authors in answering the reviewers' comments are appreciated. Based on the authors' responses, I increased the score to 6.

---

### Official Review · Reviewer_rBPQ · 2022-10-18

**Confidence:** 5
**Correctness:** 4
**Technical Novelty And Significance:** 3
**Empirical Novelty And Significance:** 4
**Recommendation:** 8

**Clarity, Quality, Novelty And Reproducibility:**

The paper appears to be technically sound, and the main idea is novel. Key codes are available, and the experimental details are provided such that an expert should be able to reproduce the main results.

**Strength And Weaknesses:**

Strength:

1. This paper is well-written and easy to follow.

2. The idea of the offset spike is interesting. The conclusion that the case of firing one more (or less) spike is the main reason for the degradation of SNN performance in ANN2SNN conversion is impressive.

3. The proposed framework is supported by theoretical analysis.

4. The proposed method achieves SOTA performance in ANN2SNN conversion and comparable performance as BPTT with less computational cost.

Weakness:
1. All figures are too small to follow.

2. The proposed method needs to take extra steps to acquire the remaining membrane potential, which decides the lower bound of the time steps.

3. The paper lacks ablation experiments to show that the proposed shifting strategy is optimal.

More questions:

1. The definition of offset spike (equation 8) lacks explanation. The authors should claim that they follow the QCFS framework with $\lambda=\theta$. Otherwise, it can not represent the gap between ANN and SNN output.

2. It is interesting to see that the proposed method has an iterative property to reduce conversion error. However, why are the Ratio and MSE in Table 3 not zero after multiple iterations?

3. Please add the ablation experiments to show that the proposed shifting strategy is optimal. The authors should compare it with other initialization strategies.

4. The proposed method needs to take extra steps to acquire the remaining membrane potential. Can you comment on the implementation on Neuromorphic chips?

5. I think figure 1 is under the condition of $T=L$. Am I right? How to rectify the firing rates of neurons in Figure 1c-d?

6. Please clarify the value of L and T in Figure 3.

**Summary Of The Paper:**

This paper proposes the offset spike to measure the deviation of expected spikes and actual spikes of SNNs in ANN2SNN conversion, so as to avoid evaluating different conversion errors and then eliminating these errors. The authors find that the case of firing one more (or less) spike is the main reason for the degradation of SNN performance in ANN2SNN conversion. Thus they propose an optimization method to reduce the offset spike by shifting up or down the initial membrane potential. Overall they achieve SOTA accuracy on CIFAR-10/100 and ImageNet datasets with fewer time steps.

**Summary Of The Review:**

This paper proposes the offset spike to eliminate ANN2SNN conversion error with a new perspective. I like the idea and would like to increase my score if the authors can solve my concerns.

---

> ### Author Response · Authors · 2022-11-17
> **Response to Reviewer rBPQ (Part 1/2)**
>
> Thanks for your constructive and thoughtful comments. We are encouraged that you found our paper well-written and easy to follow, our idea interesting, and our conclusion impressive. We would like to address your concerns and answer your questions in the following.
>
> >1. All figures are too small to follow.
>
> Thanks for your valuable advice. We have modified all figures in the revised paper to make it readable.
>
>
>
>
> >2. The definition of offset spike (equation 8) lacks explanation. The authors should claim that they follow the QCFS framework with $\lambda^l=\theta^l$. Otherwise, it can not represent the gap between ANN and SNN output.
>
> Thanks for pointing it out. We have rewritten Definition 1.
>
> **Definition 1**. We define **OFFSET SPIKE** $\boldsymbol{\psi}^l$ of layer $l$ as the difference between the desired total spike count $C_\text{designed}^l$ and the actual spike count $C_\text{actual}^l$ during the interval $[0, T]$, that is
> \begin{align}
>     \boldsymbol{\psi}^l=C_\text{designed}^l-C_\text{actual}^l=
>     \frac{\boldsymbol{a}^lT}{\theta^l }- \sum\limits_{t=1}^T \boldsymbol{s}^l(t),
> \end{align}
> where we set the maximum value $\lambda^l$ of output $\boldsymbol{a}^l$ in ANNs equal to the threshold $\theta^l$ in SNNs, that is, $\lambda^l=\theta^l$. Thus, $\frac{\boldsymbol{a}^l}{\theta^l }$ denotes the normalized output in ANNs, which is mapped to the firing rates of SNNs, and $C_\text{designed}^l=\frac{\boldsymbol{a}^lT}{\theta^l }$ denotes  the desired total spike count. Note that ${\psi}_i^l=\pm k$ indicates that the gap between the actual and desired firing rate of the $i$-th neuron in layer $l$ of the SNN is $k$ spikes.
>
> >3. It is interesting to see that the proposed method has an iterative property to reduce conversion error. However, why are the Ratio and MSE in Table 3 not zero after multiple iterations?
>
>
> Thanks for your insightful comments. We believe that iterating the proposed method can reduce conversion error to zero. We find that the non-zero MSE and Ratio in Table 2 are caused by the rounding of the floating-point numbers. Specifically, we carefully checked the Ratio, defined as the percentage of SNN input (output) equals ANN input (output) in each layer, to prove this, and we list the results in Table R3.
>
> **Table R3: Input/Output Ratio for each layer of an SNN with VGG-16 on CIFAR-10 dataset**
> | Condition                           | L1   | L2     | L3     | L4     | L5     | L6     | L7     | L8     | L9     | L10    | L11    | L12    | L13    | L14    | L15    |
> | ----------------------------------- | ---- | ------ | ------ | ------ | ------ | ------ | ------ | ------ | ------ | ------ | ------ | ------ | ------ | ------ | ------ |
> | Input Ratio                         | 100% | 99.99% | 99.97% | 99.90% | 99.63% | 99.24% | 98.77% | 97.84% | 97.43% | 97.28% | 97.31% | 97.52% | 97.11% | 97.88% | 97.80% |
> | Output Ratio                        | 100% | 99.99% | 99.99% | 99.99% | 99.99% | 99.99% | 99.98% | 99.95% | 99.94% | 99.90% | 99.68% | 99.68% | 99.74% | 99.85% | 99.87% |
> | Output Ratio when Input is accurate | 100% | 100%   | 100%   | 100%   | 100%   | 100%   | 100%   | 100%   | 100%   | 100%   | 100%   | 100%   | 100%   | 100%   | 100%   |
>
> We find that the Ratio of the output in layer 1 is 100%, but the Ratio of the input in layer 2 is close to 100%. Thus, the error must be caused by the floating point number precision problem in multiplication and division operations involved in the forward propagation between layer 1 and layer 2. Considering that SNNs will calculate $\sum\limits_{t=1}^T\boldsymbol{W}^l\boldsymbol{s}^{l-1}(t)/T$ but ANNs will calculate $\boldsymbol{W}^l(\sum\limits_{t=1}^T\boldsymbol{s}^{l-1}(t)/T)$ as the average input current for the l-th layer, these two corresponding inputs are not necessarily equal due to the rounding of the floating point number.
>
> We then conduct another experiment to prove that the conversion error can be reduced to zero if the rounding of the floating point number is eliminated. We force the input of spiking neurons to be the same as QCFS neurons in each layer and calculate the Ratio of the output. As shown in Table R3 (line 4), we find that the Ratio of the output in each SNN layer is 100%, which indicates that iterating the proposed method can finally reduce conversion error to zero. We have added these analyses in the Appendix.

---

> > ### Author Response · Authors · 2022-11-17
> > **Response to Reviewer rBPQ (Part 2/2)**
> >
> > >4. The paper lacks ablation experiments to show that the proposed shifting strategy is optimal. Please add the ablation experiments to show that the proposed shifting strategy is optimal. The authors should compare it with other initialization strategies.
> >
> > Thanks for your suggestion! We make a comparison among different initialization strategies on CIFAR-100 with VGG-16 structure, including random initialization, setting $v^l(0)={\theta^l}/2$~[1], using the residual membrane potential $v^l(\rho)$ of the first stage as the initial membrane potential and our proposed method. As shown in Table R4, our proposed method outperforms other initialization strategies under low time-steps, which proves the superiority of our method.
> >
> > **Table R4: Comparison of different initialization**
> > | Dataset  | Architecture | Method                              | T=1    | T=2    | T=4    | T=8    | T=16   |
> > | -------- | ------------ | ----------------------------------- | ------ | ------ | ------ | ------ | ------ |
> > | CIFAR100 | VGG-16       | Random Intialization                | -      | -      | -      | 68.03% | 74.74% |
> > | CIFAR100 | VGG-16       | $v^l(0)\leftarrow \theta^l/2$       | -      | 63.79% | 69.62% | 73.96% | 76.24% |
> > | CIFAR100 | VGG-16       | $v^l(0)\leftarrow v^l(\rho),\rho=4$ | 73.38% | 74.53% | 75.09% | 76.27% | 76.62% |
> > | CIFAR100 | VGG-16       | **Ours**($\rho=4$)                  | 74.24% | 76.03% | 76.26% | 76.52% | 76.77% |
> >
> >
> > [1] Tong Bu, Wei Fang, Jianhao Ding, PengLin Dai, Zhaofei Yu, and Tiejun Huang. Optimal ANN-SNN conversion for high-accuracy and ultra-low-latency spiking neural networks. In International Conference on Learning Representations, 2022.
> >
> >
> > >5. The proposed method needs to take extra steps to acquire the remaining membrane potential. Can you comment on the implementation on Neuromorphic chips?
> >
> > Thanks for your suggestion! In the first stage, we will calculate the optimal shifting distance of the initial membrane potential for specific neurons with conversion errors. After optimizing the initial membrane potential, we will spend T time steps implementing the test on corresponding datasets and deliver the output to the next layer. We have discussed this with some hardware experts, and they suggest that current neuromorphic chips, such as PAICORE [2], support ANN-SNN conversion and the operation of reading the membrane potential at each time-step. Therefore, the application of our proposed method on neuromorphic chips is feasible.
> >
> >
> >
> > [2] Yisong Kuang, Xiaoxin Cui, Yi Zhong, Kefei Liu, Chenglong Zou, Zhenhui Dai, Dunshan Yu, Yuan Wang, and Ru Huang. A 28-nm 0.34-pj/sop spike-based neuromorphic processor for efficient artificial neural network implementations. In IEEE International Symposium on Circuits and Systems (ISCAS), 2021: 1-5.
> >
> >
> > >6. I think figure 1 is under the condition of $T=L$. Am I right? How to rectify the firing rates of neurons in Figure 1c-d?
> >
> > Yes, all subfigures in Figure 1 satisfy the condition $T=L=4$. For Figure 1c-d, our constraint here denotes that we directly set the output $\boldsymbol{a}^{l-1}$ in layer $l-1$ of ANNs the same as the output $\boldsymbol{\phi}^{l-1}(T)$ of SNNs, that is, $\boldsymbol{a}^{l-1}=\boldsymbol{\phi}^{l-1}(T)$, and compute the offset spike with equation (7) and $\boldsymbol{a}^l=f(\boldsymbol{W}^l\boldsymbol{\phi}^{l-1}(T))$. Our main intention here is to explain that even without the possible interference of the conversion error existing before $l$-th layer, the case of firing one more or less spike ($\boldsymbol{\psi}^l=\pm 1$) is still the main reason for conversion error. Therefore, we can generally achieve high-performance and low-latency SNNs with only one iteration. We have added the details in the Appendix.
> >
> >
> > >7. Please clarify the value of L and T in Figure 3.
> >
> > In Figure 3, we set $L=T=4$ for both CIFAR10/VGG16 and CIFAR100/VGG16.

---

> > > ### Comment · Reviewer_rBPQ · 2022-11-18
> > > **Post-rebuttal**
> > >
> > > I would like to thank the authors for the detailed response! I have read all reviews and responses, and I am happy to increase my score to 8.

---

### Official Review · Reviewer_NhRe · 2022-10-22

**Confidence:** 5
**Correctness:** 3
**Technical Novelty And Significance:** 4
**Empirical Novelty And Significance:** 4
**Recommendation:** 8

**Clarity, Quality, Novelty And Reproducibility:**

The authors propose a new perspective to analyze the conversion errors from the offset spikes and develop a conversion method for ANN-converted SNN models.

**Strength And Weaknesses:**

Strength:
The writing is well-written and readable in general. The overall idea makes sense and demonstrates a high performance compared with other conversion methods.

Weakness:
The proposed approach requires the computation of additional offset spikes iteratively for each layer. Did the authors evaluate the additional computational cost of this method, e.g., the additional running time and the number of operations? Also, it would be better to provide a comprehensive comparison that incorporates this part with other conversion methods.

The key observation of using offset spikes for reducing conversion errors is derived from the QCFS function and IF model. Can it be generated to any conversion model with other activations or other neuron models?

The comparison between the computational complexity of backpropagation approaches and ANN-SNN conversion is unclear and inaccurate. What’s the exact meaning of complexity? Does the comparison incorporate the training of ANN parts for converted models?

Please provide necessary explanations for each figure in the corresponding legends, for example, the different types of bars in Fig. 3. Also, please adjust the font size in each figure to make it readable.

**Summary Of The Paper:**

This work proposes a spike calibration algorithm for reducing the conversion errors in ANN-SNN conversion models. The authors provide a new measurement for the conversion error by the offset spike and develop an optimization strategy that can shift initial membrane potential to offset the conversion errors iteratively. They also analyse the optimal shifting strategy theoretically. Their results on CIFAR10, CIFAR100 and ImageNet demonstrate a high inference accuracy with low latency. This paper proposes a simple and powerful method for the ANN-SNN conversion model, which promises in deploying the neuromorphic chips for energy-efficient inference.

**Summary Of The Review:**

This paper proposes a simple and powerful method for the ANN-SNN conversion model, which promises in deploying the neuromorphic chips for energy-efficient inference.

---

> ### Author Response · Authors · 2022-11-17
> **Response to Reviewer NhRe (Part 1/2)**
>
> Thanks for your insightful and valuable comments. We are encouraged you find that our paper is well-written, and our idea makes sense and demonstrates a high performance compared with other conversion methods. We would like to address your concerns and answer your questions in the following.
>
> >1. Did the authors evaluate the additional computational cost of this method, e.g., the additional running time and the number of operations? Also, it would be better to provide a comprehensive comparison that incorporates this part with other conversion methods.
>
> Thanks for your insightful comments. Previous works usually adopt the "step-by-step" mode, which means that the input is transmitted in each layer of the network at each time step. For our works, as we need to provide corresponding accurate input for each layer to eliminate the conversion error (offset spike) effectively, we adopt the "layer-by-layer" mode, which means that the output is calibrated and tested for $\rho+T$ steps on this layer before being delivered to the next layer. In general, the only difference between the two modes is that they exchange the operation order of time-steps and network layers.
>
> As we adopt the optimal shifting distance proposed in Theorem 2 to obtain better performance, the maximum/minimum operation involved will increase our method's memory and time cost. However, as suggested by Reviewer rBPQ, we can also provide a lightweight optimization scheme: for each layer, we can consider directly selecting the residual membrane potential $\boldsymbol{v}^l(\rho)$ after $\rho$ steps as the initial membrane potential for our second stage. The idea is to make $\boldsymbol{v}^l(T)-\boldsymbol{v}^l(0)$ in equation (5) approach 0 to eliminate the conversion error (offset spike). Table R1 reports the results on the ImageNet dataset. Although the performance of our lightweight optimization scheme is weaker than our best solution, it is still much better than the current SOTA methods. Under the condition of using the lightweight scheme, we can avoid the extra calculation of the optimal shifting distance. As you suggested, we compare the running time among QCFS, our lightweight scheme, and our shifting method on CIFAR-100 with VGG-16 structure and 16 time-steps. The corresponding running time is 101s, 101s, and 134s, respectively. We have added the comparsion in the Appendix.
>
> **Table R1: Comparison with the state-of-the-art ANN-SNN conversion method**
> | Dataset  | Architecture | Method      | T=1    | T=2    | T=4    | T=8    | T=16   |
> | -------- | ------------ | ----------- | ------ | ------ | ------ | ------ | ------ |
> | ImageNet | VGG-16       | OPI | - | - | - | 6.25% | 36.02% |
> | ImageNet | VGG-16       | QCFS | - | - | - | 19.12% | 50.97% |
> | ImageNet | VGG-16       | lightweight(ours) | 62.27% | 69.69% | 72.50% | 73.39% | 74.04% |
> | ImageNet | VGG-16       | shifting(ours)    | 63.84% | 70.59% | 72.94% | 73.82% | 74.09% |
> | ImageNet | ResNet-34       | QCFS | - | - | - | 35.06% | 59.35% |
> | ImageNet | ResNet-34    | lightweight(ours) | 69.04% | 69.63% | 69.80% | 69.77% | 70.97% |
> | ImageNet | ResNet-34    | shifting(ours)    | 69.11% | 72.66% | 73.81% | 74.17% | 74.14% |

---

> > ### Author Response · Authors · 2022-11-17
> > **Response to Reviewer NhRe (Part 2/2)**
> >
> > >2. The key observation of using offset spikes for reducing conversion errors is derived from the QCFS function and IF model. Can it be generated to any conversion model with other activations or other neuron models?
> >
> > Yes, it also applies to other conversion models. Our theorems and mathematical derivation are based on the QCFS-IF framework, but we can also obtain remarkable improvement on other conversion models with different activations, including TCL[1] and OPI[2], which have been shown in Table R2. As most of the previous works in ANN-SNN conversion are all based on IF neuron models with the "reset-by-subtraction" mechanism, the mainstream activation functions, including QCFS, are not applicable to other neuron models (such as LIF, SRM, and HH), which deserves further research.
> >
> >
> > **Table R2: Comparison about the optimization of other activation layer**
> > | Dataset     | Architecture     | Method       | T=4     | T=8     | T=16    | T=32    | T=64     |
> > | ----------- | ---------------- | ------------ | ------- | ------- | ------- | ------- | -------- |
> > | CIFAR100    | VGG-16           | TCL          | -       | -       | 17.94%  | 52.30%  | 71.17%   |
> > | CIFAR100    | VGG-16           | **TCL+Ours** | 63.53%  | 74.69%  | 75.05%  | 75.82%  | 76.37%   |
> > | **Dataset** | **Architecture** | **Method**   | **T=1** | **T=2** | **T=4** | **T=8** | **T=16** |
> > | CIFAR100    | VGG-16           | OPI          | -       | -       | -       | 60.49%  | 70.72%   |
> > | CIFAR100    | VGG-16           | **OPI+Ours** | 71.20%  | 73.45%  | 73.42%  | 73.98%  | 75.23%   |
> >
> > [1] Nguyen-Dong Ho, Ik-Joon Chang. TCL: an ANN-to-SNN conversion with trainable clipping layers. The 58th ACM/IEEE Design Automation Conference (DAC), 2021: 793-798.
> >
> > [2] Tong Bu, Jianhao Ding, Zhaofei Yu, Tiejun Huang. Optimized potential initialization for low-latency spiking neural networks. The Thirty-Sixth AAAI Conference on Artificial Intelligence (AAAI), 2022: 11-20.
> >
> >
> > >3. The comparison between the computational complexity of backpropagation approaches and ANN-SNN conversion is unclear and inaccurate. What’s the exact meaning of complexity? Does the comparison incorporate the training of ANN parts for converted models?
> >
> > Yes, our comparison incorporates the training of ANN parts for converted models. The comparison between the computational complexity of BPTT and ANN-SNN conversion here is mainly about the memory resource during the network training procedure. For ANN-SNN conversion, we first train an ANN with the QCFS function, which only involves the back-propagation from the back layer to the front layer. However, for BPTT, back-propagation includes not only the propagation between layers, but also the propagation from the next time step to the previous time step for the spiking neurons in each layer, which increases the memory resource. In the revised paper, we change it to "Note that compared to ANN-SNN conversion, the back-propagation approaches need to propagate the gradient through both spatial and temporal domains during the training process, which consumes large amounts of memory and computing resources".
> >
> >
> >
> > >4. Please provide necessary explanations for each figure in the corresponding legends, for example, the different types of bars in Fig. 3. Also, please adjust the font size in each figure to make it readable.
> >
> > Thanks for your suggestion! We have adjusted the corresponding legends and the font size of all figures in the revised paper. For Fig. 1, "w/ constraint" denotes that we set the output $\boldsymbol{a}^{l-1}$ in layer $l-1$ of ANNs the same as the output $\boldsymbol{\phi}^{l-1}(T)$ of SNNs, that is, $\boldsymbol{a}^{l-1}=\boldsymbol{\phi}^{l-1}(T)$, and "w/o constraint" stands for not performing the above additional operations. For Fig. 3, "w/o shift" stands for not using the optimization method, "w/ shift" denotes using the optimization method for one time. For Fig. 5 and Fig. 6, "shift $\times$ k" represents iteratively using the optimization method for k times. For Fig. 1, Fig. 3, and Fig. 6, $\psi^l=\pm k$ denotes the specific value of the offset spike according to Definition 1.

---

> > ### Comment · Reviewer_NhRe · 2022-11-23
> > **Response to rebuttal**
> >
> > Thanks for the authors' nice responses. My concerns have been addressed.

---

### Official Review · Reviewer_oG9M · 2022-10-28

**Confidence:** 4
**Correctness:** 3
**Technical Novelty And Significance:** 3
**Empirical Novelty And Significance:** 3
**Recommendation:** 6

**Clarity, Quality, Novelty And Reproducibility:**

The paper has significant problems concerning readability.
A lot of spelling and grammar mistakes make the text incomprehensible, and in many occasions, the reader has to guess what a sentence means.

"low time-steps" means low number of time-steps or steps of short time?

"and answer the question of where the extreme performance of ANN-SNN conversion is."
probably means which are the reasons in the lack of performance of ANN-SNN conversion methods.

The text is rich in many such examples.



**Strength And Weaknesses:**

+: The paper has a good introduction to the ANN-SNN conversion paradigm
+: The authors make two good observations about the offset spike
+: The authors state and show two theorems. While the math seems right, it is difficult to judge their relevance

Unfortunately, the paper is pretty much unreadable and I could only understand the technical part following the equations only but not the text.


**Summary Of The Paper:**

The authors address the issues underlying ANN to SNN conversion and identify the firing of one more or less spikes as the main reason of failing conversion. This can be exploited in an optimization procedure over existence of an offset spike. Evaluation is done on simulated firing rates of cifar-100.


**Summary Of The Review:**

The paper has very good technical content and it would make an acceptable contribution if the authors would write the non-mathematical part from scratch.
I apologize that I cannot give a longer review, but the paper is not comprehensible.

Authors have made a significant revision and their efforts are really appreciated.

---

> ### Author Response · Authors · 2022-11-17
> **Response to Reviewer oG9M**
>
> Thanks for your constructive and valuable feedback. We are encouraged that you find our paper with a good introduction to ANN-SNN conversion and very good technical content. We would like to address your concerns and your questions in the following.
>
> >1. The paper is pretty much unreadable and I could only understand the technical part following the equations only but not the text. In many occasions, the reader has to guess what a sentence means.
>
> Thank you for your valuable suggestion! We have uploaded our revised version, where we thoroughly checked and corrected the spelling and grammar mistakes, and added more explanation to the theorems, and clarified the algorithm with an additional diagram. We hope that our modifications improve the readability of the paper, and we sincerely look forward to your further evaluation of our work.
>
> >2. "low time-steps" means low number of time-steps or steps of short time?
>
> Thanks for pointing it out! The concept "low time-steps" here mainly denotes a low number of time-steps. For each time-step, the corresponding input from the dataset will pass through the entire SNN network layer by layer, and spiking neurons in each layer will deliver binary spikes. We will count the average spike firing rate within a period and view the corresponding subscript of the neuron with the highest firing rate as the label of network prediction. In previous works, SNNs have always been challenging to obtain high accuracy on large-scale datasets under the condition of a low number of time-steps. We proposed an optimization strategy based on shifting up or down the initial membrane potential, which remarkably improves the performance of SNNs under a low number of time-steps.
>
> >3. "and answer the question of where the extreme performance of ANN-SNN conversion is." probably means which are the reasons in the lack of performance of ANN-SNN conversion methods.
>
> Thanks for your insightful comments. We wrote this sentence initially to express two meanings. One is to understand, as you have commented, why there is a performance gap between the converted SNN and the source ANN. Another meaning is to ask, can we get an upper bound of the conversion performance (we address this by proposing iterative variants of our method)? In light of your comments, we change the sentence to: "answer the question of how to improve the performance of a converted SNN and possibly approach the upper bound  performance".

---

> ### Author Response · Authors · 2022-11-30
> **Thank you for the time and hope our responses helpful for your re-assessment of our work.**
>
> Dear reviewer oG9M,
>
> Thank you for the thorough feedbacks and constructive suggestions. We sincerely hope our posted response and revised version can help to address your concerns on our paper and serve as a reference for your re-assessment of our work. If you have any further comments and questions, please let us know and we are glad to write a follow-up response.

---

### Decision · Program_Chairs · 2023-01-20

**Decision:**

Accept: poster

**Justification For Why Not Higher Score:**

Clarity issues with the work.  The reviewers all found the figures hard to understand, which implies to me that it wouldn't make for a good presentation.

**Justification For Why Not Lower Score:**

All the reviewers voted for accept.  The work seems timely, appropriate and developing methods to enable lower-power hardware for deep neural nets seems important.

**Metareview: Summary, Strengths And Weaknesses:**

This paper addresses the conversion of standard artificial neural networks, e.g. deep nets, into spiking neural network formulations which are more biologically plausible and can be more compute / power efficient in specialized hardware.  This paper analyses where errors originate from in the conversion process and develop a calibration method to minimize the conversion error.  They empirically evaluate on a variety of common image benchmark problems and achieve strong results for SNNs.

One reviewer found the work poorly written and difficult to understand, while curiously others remarked that is was well-written and easy to follow.  However, three reviewers found the figures too small and difficult to interpret and multiple reviewers found the presentation of the offset spike difficult to understand.  The reviewers agreed that the paper was well-motivated, the method seemed sensible and the empirical evaluation was convincing.  One reviewer found that the computational complexity of computing the offset spike wasn't adequately addressed in the work.

Overall, the reviewers agreed that the paper should be accepted (6, 6, 8, 8).  The work seems timely, appropriate for ICLR and interesting to the community.  Therefore the recommendation is to accept the paper.

**Note From Pc:**

if the above contains the word "oral" or "spotlight" please see: "oral" presentation means -> notable-top-5% and "spotlight" means -> notable-top-25%. As stated in our emails, we are disassociating presentation type from AC recommendations